# Diel and eddy driven changes in microbial gene expression and biogeochemistry in the oceanic chlorophyll maximum

Logan M. Peoples [1] ✉, John M. Eppley [2], Benedetto Barone [2], Brett W. Hobson[3], David M. Karl [2], Brian Kieft[3], Roman Marin III[3], Christina M. Preston[3], Anna E. Romano[2,4], John P. Ryan[3], Christopher A. Scholin [3], Samuel T. Wilson[2,5], Yanwu Zhang[3], Matthew J. Church [1] ✉ & Edward F. DeLong [2] ✉

Oceanic microorganisms can rapidly respond to environmental variability. Determining how physical and biological processes control microbial distributions, abundances, and metabolic dynamics is challenging. Here, we used autonomous underwater vehicles capable of Lagrangian feature tracking and in situ sampling, in combination with ship-based measurements, to examine diel to weekly scale changes in microbial transcription and biogeochemistry in the deep chlorophyll maximum (DCM) of a mesoscale cyclonic eddy. Nearly 20% of total transcript expression showed diel periodicity, highlighting the importance of the diurnal cycle on phytoplankton metabolism in the dim waters of the DCM. Eddy-induced isopycnal uplift increased nutrient concentrations and caused upward displacement of the DCM, driving increased picoeukaryotic cell abundances and transcriptional activity of nitrate-incorporating photoautotrophs. As the eddy weakened, the DCM deepened and transcriptional activity shifted towards chemolithoautotrophic ammonia-oxidizing archaea. The temporal dynamics observed demonstrate how plankton communities rapidly respond to both diel variation and stochastic mesoscale disturbances.

Photosynthetic harvesting of sunlight by microscopic plankton in the ocean fuels nearly half of the annual primary production on Earth[1,2]. Organic carbon produced by phytoplankton supports tightly connected oceanic food webs that consume most primary production in the sunlit layer of the upper ocean (i.e., the photic zone). The exponential decrease of light energy through the upper ocean structures strong vertical gradients in photosynthesis, plankton populations, and nutrient concentrations[3–5]. On daily to seasonal scales, variations in solar energy input influence the timing of plankton cell division[6,7], gene expression[8–10], and metabolite and nutrient uptake[11–13].

Plankton are closely connected to an array of heterogeneous physical features that define the ocean habitat[14]. Microorganism growth and physiology can respond rapidly to variations in energetic and nutritional resources triggered by these physical features. Most kinetic energy in the ocean is found in meso- and submesoscale physical events, including turbulent fronts, eddies, and waves, which span tens to hundreds of kilometers and persist for weeks to months[15,16]. These features can alter biological and biogeochemical processes by modifying the vertical structure of ocean ecosystems[17–19]. Eddies are rotating bodies of water that propagate through the upper ocean and

[1]Flathead Lake Biological Station, University of Montana, Polson, MT, USA. [2]University of Hawai'i at Mānoa, Honolulu, HI, USA. [3]Monterey Bay Aquarium Research Institute, Moss Landing, CA, USA. [4]California Institute of Technology, Pasadena, CA, USA. [5]Newcastle University, Newcastle upon Tyne, UK. ✉e-mail: logan.peoples@flbs.umt.edu; matt.church@flbs.umt.edu; edelong@hawaii.edu

vertically displace isopycnals across the strong photic zone light gradient, causing profound impacts on nutrient distributions and plankton physiology. Eddies are present across one-third of the ocean surface area in the mid-latitudes[15,20]. In the northern hemisphere, cyclonic eddies, characterized by negative sea level anomalies (SLA) and isopycnal uplift, can displace nutrient-enriched waters into the photic zone, accelerating primary productivity, increasing plankton biomass, and altering microbial community composition[21-23]. Because eddies can persist for months, have varying intensities, and exhibit changes in coherency (i.e., mixing with external water), their impact on microbial community composition and activity likely changes as these features mature[24-26].

In the oligotrophic North Pacific Subtropical Gyre (NPSG), the photic zone is often conceptualized as a two-layer system: an upper layer (0–50 m) with high light flux but persistently low bioavailable nutrients above a lower layer (100–175 m) where light limits photosynthesis and nutrient concentrations increase with depth[27-29]. The lower layer includes the deep chlorophyll maximum (DCM), a feature ubiquitous in the subtropical ocean gyres, where chlorophyll concentrations increase as light energy decreases to <1% surface light flux. The DCM reflects a confluence of resource gradients, where decreasing light flux from above meets increasing nutrient concentrations that enable phytoplankton growth[30,31]. While diel-scale variations in microbial physiology and gene expression are evident across diverse phyla in the well-lit photic zone, whether microorganisms in the dimly lit DCM also demonstrate physiological and transcriptional responses to daily fluctuations in light energy remains less clear[32]. Motion associated with internal waves that propagate through the ocean pycnocline can cause vertical oscillations and displacement of the DCM by tens of meters over short time scales, making tracking of temporal variations in microbial populations and metabolism at the base of the photic zone challenging.

Here, we explored the spatiotemporal mosaic of microbial activities at the DCM and its response to a mesoscale cyclonic eddy across multiple temporal scales. Autonomous Underwater Vehicles (AUVs) capable of identifying, tracking, and Lagrangian sampling of the DCM, along with classical ship-based sampling, were used to study microbial populations over time. By sampling at high temporal resolution (hourly) for nearly three weeks, we sought to answer the following questions: 1) How does eddy isopycnal displacement impact microbial communities in the DCM? 2) To what extent do microorganisms in the DCM show evidence of diel periodicity in gene expression? and 3) How does microbial population structure and physiology change as an eddy weakens? We contextualize this short-term variability through comparisons against the long-term climatology (1988-present) at the nearby research site Station ALOHA[4]. We find that microbial populations respond to changes in both diurnal variations in light and mesoscale eddy strength, emphasizing how different time and space scales influence plankton physiology and biogeochemistry in the ocean.

## Results

### Isopycnal uplift stimulates phototrophs at the DCM

This study targeted a strong cyclonic eddy north of the Hawaiian Islands during March–April 2018 (Fig. 1A). Hindcasting suggests eddy genesis occurred in early February 2018 with the eddy moving southward while intensifying to its maximum strength in March (as inferred by changes in SLA, approximately −18 cm; Supplemental Fig. 1). Based on the historical SLA from Station ALOHA, located ~200 km away, this was an anomalously strong event for the region (Fig. 1D). The cyclone varied between ~50 and 90 km in radius with the core remaining relatively coherent during the period of our sampling, suggesting limited mixing of microbial populations with waters outside the eddy[20,33].

To examine plankton physiological and community compositional responses to isopycnal uplift at diel and mesoscale resolution, a suite of Lagrangian instrumentation and shipboard-based CTD casts

were used to sample the DCM for over 3 weeks during two cruise legs (Leg 1, March 10–24; Leg 2, March 28-April 4). This instrumentation included two AUVs that dynamically tracked and sampled the DCM: AUV *Opah* spiraled up and down the water column, measuring hydrographic properties, while AUV *Aku* followed and sampled an isotherm corresponding to the depth of the peak of chlorophyll fluorescence (i.e., the DCM;[34,35]). Continuous AUV sampling occurred over two ~3-day periods (Fig. 1B, Supplemental Figs. 2–4). On time scales of a few hours, high-frequency variability in internal-wave-driven isopycnal fluctuations resulted in oscillations in the vertical position of the DCM by more than 30 m. Despite these large vertical excursions, *Aku* successfully tracked the DCM with high precision, falling within 0.15 °C of the targeted isotherm 98% of the time.

The position, biogeochemistry, and microbial community of the DCM were impacted by the cyclonic eddy. The depth of the DCM averaged 102 m, approximately 15 m shallower than typical (117 m) for Station ALOHA and the NPSG. Moreover, the DCM was centered around the 24.7 kg m$^{-3}$ isopycnal (range across both cruise legs was ~24.6 to 24.8 kg m$^{-3}$), a density surface whose mean vertical position resides near 151 m at Station ALOHA, consistent with isopycnal uplift of ~50 m associated with this eddy (Fig. 1E). Although isopycnal uplift introduced waters with elevated concentrations of N + N and phosphate to the lower photic zone, comparison to the Station ALOHA nutrient climatology revealed concentrations that were lower than typical for this isopycnal (24.7 kg m$^{-3}$; Fig. 1F, Supplemental Fig. 5A–C). These observations are consistent with prior biological assimilation of nutrients due to the uplift of isopycnals into the photic zone. Elevated chlorophyll concentrations and cell abundances of photosynthetic picoeukaryotes, in comparison to typical conditions at both the DCM and the 24.7 kg m$^{-3}$ isopycnal, indicate that the upward displacement of nutrient-enriched water stimulated primary production (Fig. 1G–J, Supplemental Fig. 5D). While neither *Prochlorococcus* nor heterotrophic picoplankton cell abundances were enriched throughout the water column, both picoplankton groups were 4× more abundant than typical for the 24.7 kg m$^{-3}$ isopycnal (Fig. 1K, Supplemental Fig. 5E–G), consistent with a response to isopycnal uplift.

### Photoautotrophs are the most transcriptionally active lineages at the DCM

Quantitative metatranscriptomic sequencing of the DCM was performed on 97 samples, 61 from AUV *Aku* and 36 collected using a ship-based CTD rosette (Supplementary Data 1). We mostly focus on results from AUV samples that were collected every ~3 h (see Supplemental Information). Cyanobacteria were the most transcriptionally active lineage at the DCM (Fig. 2, Supplemental Fig. 6). At the genus level, *Prochlorococcus*_B (Low Light clade I, LLI), which is known to be abundant in the lower euphotic zone[36-38], and *Prochlorococcus*_A (HL, High Light clade) were the most active Cyanobacteria (average relative expression of 11.3% and 1.9%, respectively). The Marine Group II archaea (MGII, within the phylum Thermoplasmatota or Euryarchaeota), which are often present at the DCM[39,40], represented on average 8% of each metatranscriptome and were largely represented by the class Poseidoniia, including the genera MGIIb-O3 and MGIIb-O5 (3.6% and 3.0%, respectively). The genus *Pelagibacter* within the SAR11 clade (5.4%) and AG-430-B22 within the OCS116 clade (2.0%) in the Alphaproteobacteria were also some of the most transcriptionally active lineages. Transcripts identified as belonging to the Eukaryota represented ~6% of each sample, with the most abundant group being *Pelagomonas* (2%) and smaller contributions from *Bathycoccus* (0.4%), *Chrysochromulina* (0.4%), *Ostreococcus* (0.3%), MAST-4A (0.2%), and *Prymnesium* (0.2%). Unclassified transcripts averaged 39% of the total transcript expression per sample. When considering the diversity of transcripts (total number of unique transcripts) within each sample rather than their level of expression, the genera *Pelagibacter*

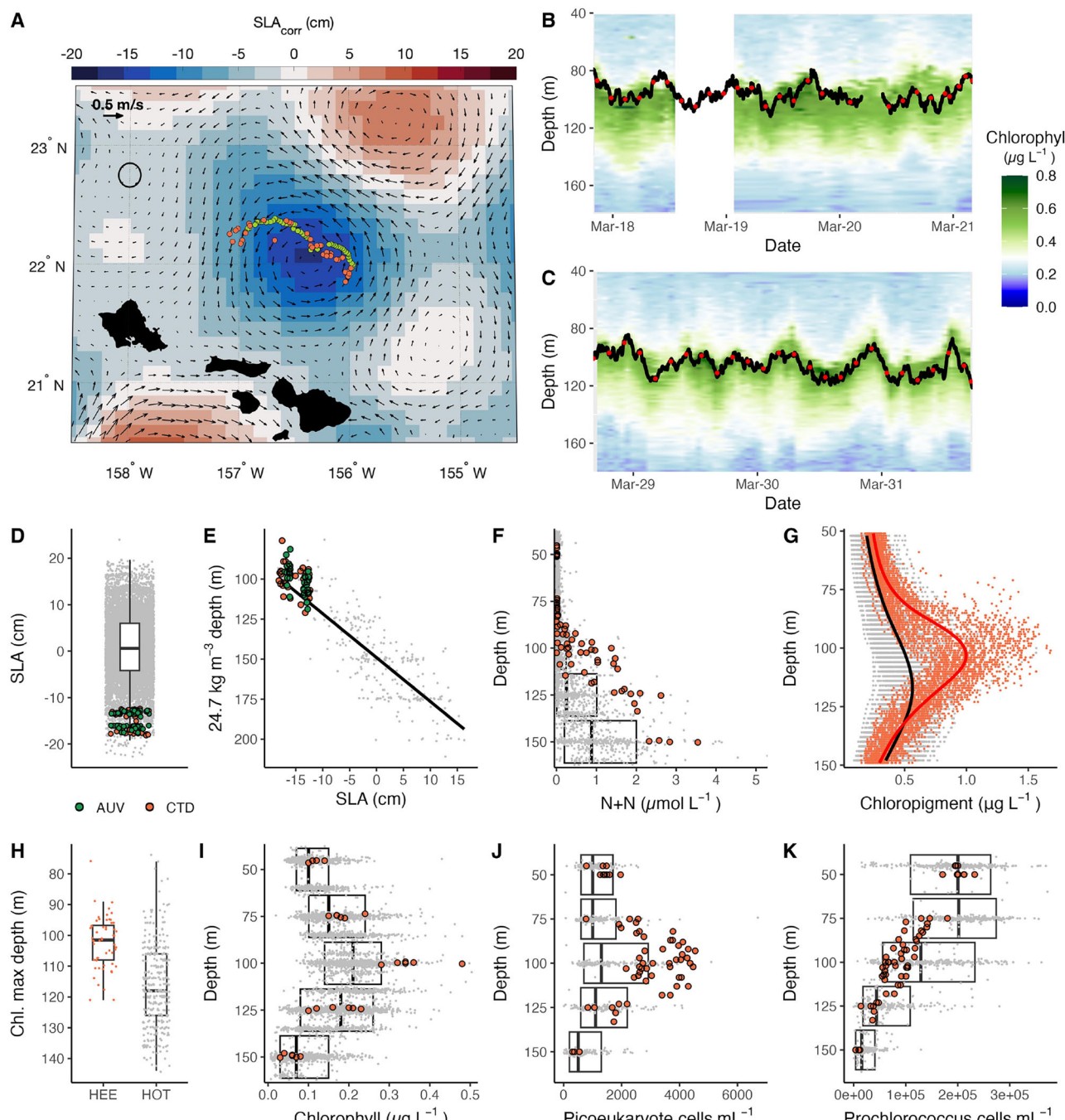

**Fig. 1 | Shipboard and Autonomous Underwater Vehicle (AUV) sampling during the Hawaiian Eddy Experiment (HEE) captured a physically and biogeochemically intense cyclonic eddy. A** Map of the sampling locations and sea level anomaly (SLA; March 25, 2018) during this study. Green points represent AUV and orange shipboard CTD sampling locations. The black circle reflects the location of the long-term research location Station ALOHA. Arrows reflect the direction and strength of absolute geostrophic currents. **B, C** Representative chlorophyll fluorescence profiles collected by AUV *Opah* during part of Leg 1 (**B**) and Leg 2 (**C**), overlayed with the position of AUV *Aku* (black line) and transcriptome sampling time points (red points). AUV-derived chlorophyll concentrations were unavailable when AUV *Opah* periodically surfaced (white regions in panel (**B**)). **D** SLA at *Aku* and shipboard CTD rosette sampling stations relative to daily SLA values from the Hawaii Ocean Time-series (HOT) at Station ALOHA (years 1993–2019). For all following panels, AUV samples are shown in green, CTD rosette sampling in orange, and HOT program observations in gray. **E** The depth of DCM sampling points (~ 24.6–24.8 kg m$^{-3}$) compared to typical depths of the 24.7 kg m$^{-3}$ isopycnal, (**F**) nitrate+nitrite (N + N) concentrations, (**G**) chloropigment depth profiles, (**H**) the depth of the chlorophyll maximum, (**I**) discrete fluorometric chlorophyll *a* concentrations, and (**J**) picoeukaryote and (**K**) *Prochlorococcus* cell abundances during the HEE expedition relative to historical HOT measurements at Station ALOHA. Boxplots throughout reflect the median, 25th, and 75th percentiles of HOT data. The line in (**E**) is fit using a linear model, while those in (**G**) are fit using a loess function.

(average 6.7%), *Prochlorococcus*_B (2.1%), and *Prochlorococcus*_A (2.1%) expressed the highest number of unique transcripts.

Next, we explored functional activity at the DCM (Fig. 2). The most highly expressed KEGG Ortholog (KO) categories across the entire dataset reflected the importance of photosynthesis and carbon

fixation, cell division and replication, and the uptake of both organic and inorganic compounds, especially nitrogen in the form of ammonium, proteins, and amino acids (Supplementary Data 2). *Prochlorococcus*_B and members of the Eukaryota, including *Pelagomonas*, were most active in expression of genes related to photosynthesis,

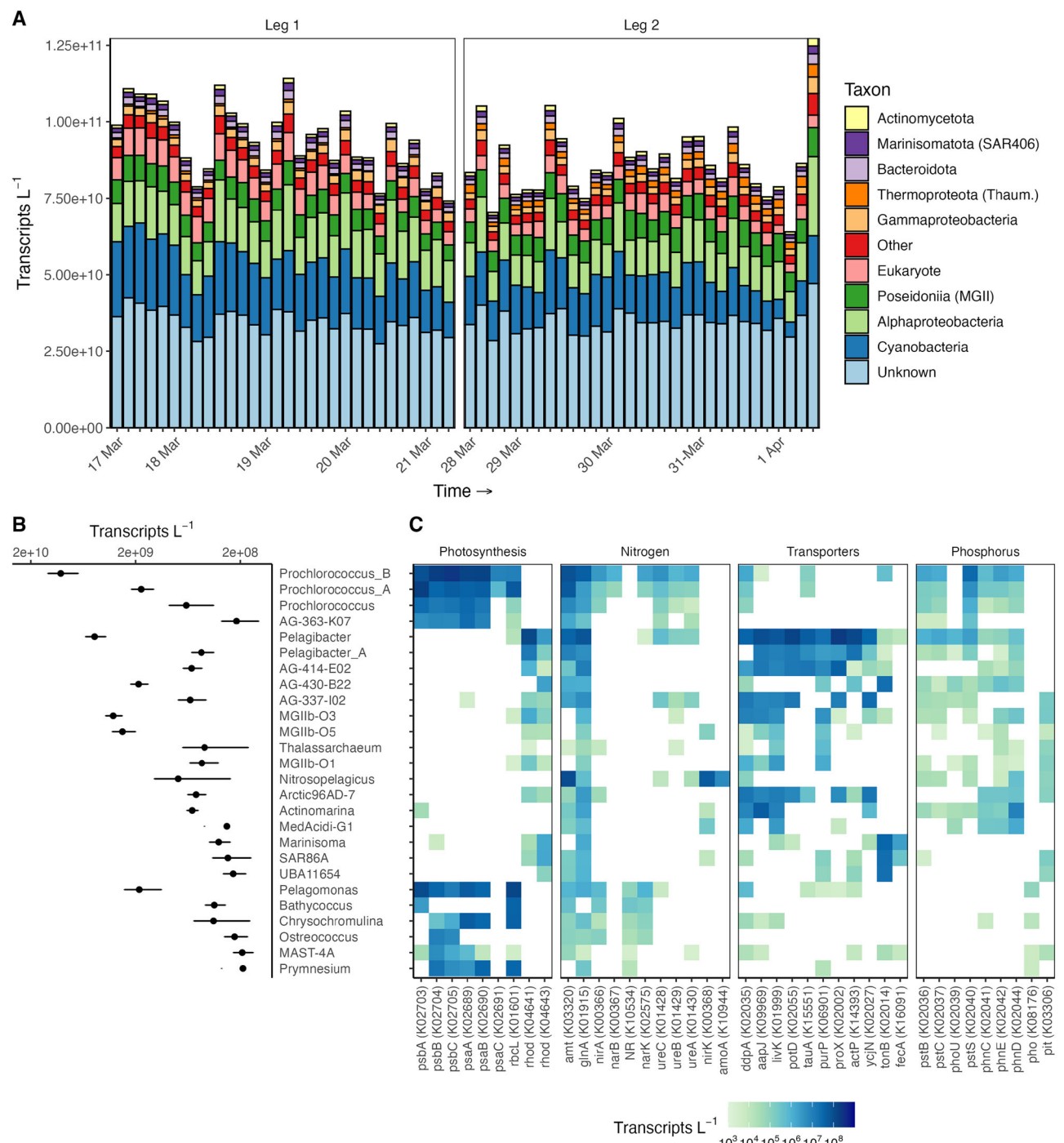

**Fig. 2 | Photosynthetic organisms are some of the most transcriptionally active groups at the deep chlorophyll maximum layer (DCM). A** Absolute transcript abundance (transcripts $L^{-1}$) of abundant taxonomic groups sampled by AUV at the DCM. Sample names are removed for clarity, but are ordered by time of collection from left to right, with the first sample collected on each day labeled. Leg 1 samples were obtained when the eddy was stronger, while Leg 2 samples were obtained as the eddy began to weaken. **B** The average total transcripts $L^{-1}$ per sample of abundant genus-level taxonomic groups and (**C**) their expression of certain KEGG ortholog (KO) functional transcripts collected by AUV. Error bars reflect the standard deviation.

while the cyanobacteria dominated expression of genes involved in nitrogen uptake, particularly ammonium and urea. Reductases for the assimilation of nitrate and nitrite were evident among both cyanobacteria and eukaryotes, including in the LLI clade *Prochlorococcus*_B (Fig. 2, Supplemental Fig. 7). Heterotrophic lineages demonstrated high expression of genes involved in the uptake of organic compounds. For example, transcripts specific to transport of glycine betaine/proline (KO K02002), spermidine/putrescine (K02055), ammonium (K03320), taurine (K15551), acetate (K14393), and amino acids (K09969, K01999, K02035) were highly expressed by *Pelagibacter*, consistent with these compounds comprising important sources of carbon, nitrogen, and energy for these microorganisms[41–43]. Based on transcriptional patterns in the DCM, *Pelagibacter* appears to couple transport of these compounds to phototrophic energy generation[44,45]: up to 50% of total putative rhodopsin gene expression was linked to the Pelagibacterales (Supplemental Fig. 8). Many of the most highly expressed transcripts across the entire dataset were hypothetical proteins putatively related to the Poseidoniales within the

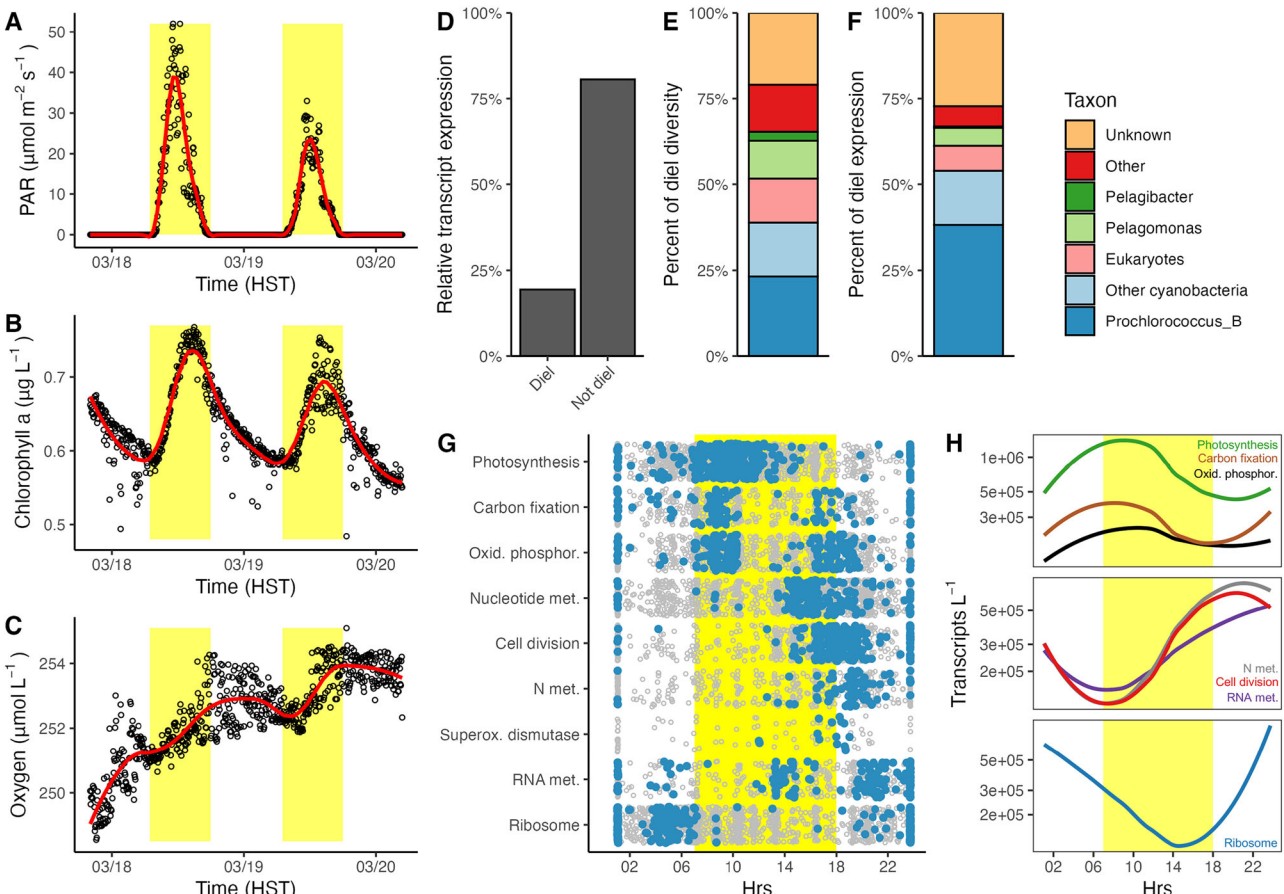

**Fig. 3 | Diel patterns in physiology and transcript expression are evident at the DCM. A–C** Representative flux of photosynthetically active radiation (PAR), chlorophyll a fluorescence, and oxygen ($O_2$) concentrations measured by AUV *Aku* at the DCM during the first leg. Yellow regions depict daylight periods in Fig. 3. **D** The percent of total transcript expression, summed across the entire dataset, that demonstrates diel periodicity. **E** The taxonomy of AUV diel-expressed transcripts when considering either just their diversity or (**F**) their summed total expression across the entire dataset. Colors are the same in (**E**, **F**). **G** The time of maximum expression of transcripts within the Cyanobacteria is organized by functional grouping. Transcripts in blue show evidence of diel expression, while those in gray do not. See the Supplemental Information for a complete explanation of transcript functional groupings. **H** Average hourly expression of cyanobacterial transcripts identified as showing diel rhythmicity in (**G**), colored by function and organized by time of maximum expression. Met, metabolism.

MGII archaea; this group also expressed transcripts involved in the uptake and degradation of amino acids and proteins (pseudolysins, K01399; serine proteases, K17734; uptake, K01999, K09969, K02035, K11954), growth and cell division (K13798, K03041, K06901, K03124, K03234, K03042), ATP synthesis (K03924, K02117, K02118), and the production of bacteriorhodopsins (data not shown). Other abundant groups and their functions included ammonia oxidation (K10944) within *Nitrosopelagicus* (phylum Thaumarchaeota or Thermoproteota; Fig. 2) and tonB-dependent receptors (K02014) within the AG-430-B22 (Alphaproteobacteria), Marinisoma (Marinisomatota, previously Marinimicrobia or SAR406) and SAR86A (class SAR86, Gammaproteobacteria; Fig. 2, Supplemental Fig. 8).

### Physiological and transcriptional rhythms are evident at the DCM

While gene expression and metabolism fluctuate on diel scales in the surface ocean, it remains unclear whether such patterns persist in the low-light regions of the lower photic zone. Here, both AUV oxygen and fluorescence-derived chlorophyll concentrations at the DCM exhibited diel periodicity during the first sampling leg (Fig. 3, Supplemental Fig. 9). Chlorophyll concentrations peaked in the midafternoon (~15:00 h) and then declined through the night, a pattern consistent with accumulation of biomass during the day and removal at night (~18:00–07:00). Dissolved oxygen concentrations also increased

during daylight hours, reaching a maximum in the early evening (~18:00) before declining through the night (minimum near ~07:00), consistent with net photosynthetic production during the day and respiratory consumption at night. Coinciding with these daily changes, we identified 13,735 transcripts that exhibited significant diel oscillations, with 3449 of these showing periodicity during both sampling legs (Fig. 3, Supplementary Data 3). Rhythmic transcripts represented 0.3% of all unique transcripts (>4 million), 3.8% of the transcripts screened (those present in >10 samples, ~358,000), and 4.2% the top 100,000 most expressed transcripts (4159 unique transcripts). When considering the expression level of transcripts by summing the quantitative abundances of all diel-expressed transcripts identified on either leg relative to the expression of all transcripts across the entire dataset, diel-expressed transcripts comprised 19.4% of total expressed transcript abundance, highlighting the importance of diel variability in transcription even in the poorly lit waters of the DCM. Transcripts from nearly all functional categories demonstrated diel variability, particularly genes associated with photosynthesis, carbon fixation, ATP synthesis, nutrient transport, cell division, and ribosomal protein synthesis. More than 50% of the diel transcripts, both in terms of diversity and overall expression, were derived from photosynthetic cyanobacteria and eukaryotic lineages. Within the cyanobacteria, genes involved in photosynthesis, carbon fixation, and oxidative phosphorylation were highly expressed in the early morning hours

( ~ 07:00–10:00), with transcripts putatively involved in the uptake of iron becoming increasingly abundant by midday, a finding consistent with the role of iron in photosynthetic energy-transferring proteins[46]. By the afternoon and evening (14:00–19:00), cyanobacterial transcripts involved in nucleotide metabolism (purine and pyrimidine), cell division, and nitrogen metabolism and uptake (i.e., ammonium transporters, glutamine synthetase) became highly expressed. Finally, transcripts involved in ribosomal protein synthesis were most highly expressed through the night and early morning. Similar results were observed using weighted gene network correlation analysis (Supplemental Fig. 10). While non-photosynthetic lineages represented a small proportion of transcripts that exhibited diel rhythms (**Supplemental Information**), gene network correlation analysis identified a group of transcripts belonging to heterotrophic organisms that showed synchronicity but did not oscillate across a 24 h cycle (Supplemental Fig. 10F). These transcripts belonged to putatively heterotrophic lineages such as AG-430-B22 and *Pelagibacter* and included a large number of tonB-dependent receptors which are involved in the transport of siderophores, B12, metals, and carbohydrates[47], a proton pump (K15987), carbon monoxide dehydrogenase (K03520), and glutamate synthase (K00265).

### Microbial physiology responds to changes in eddy strength

While the mesoscale cyclonic eddy clearly perturbed the microbial community at the DCM relative to typical conditions in the NPSG, how did this community respond to changes in physical forcing over time? Over the ~3 week period of study, the intensity of the cyclonic eddy decreased (Fig. 4, Supplemental Figs. 1, 4), with corresponding relaxation of isopycnal uplift and a deepening of the surface mixed layer (Supplemental Fig. 2). The horizontal position of subsurface water masses sampled within the eddy also changed over time, with the AUVs sampling further from the center of the eddy during the second leg (Supplemental Fig. 4). These physical and spatial changes, in part associated with eddy weakening, had clear influences on biology (Fig. 4). The intensity of the DCM weakened and the feature deepened (from ~98 m to 105 m), consistent with downward displacement of isopycnals. During the 2nd cruise leg, the DCM resided along a slightly deeper but warmer isotherm, with the temperature increasing from approximately 20.4 °C to 21.0 °C. Wirewalker-measured backscatter at the DCM, a proxy for particle concentration or volume, increased between the 1st and 2nd sampling legs and then rapidly declined, suggesting rapid particle production followed by removal, likely via consumption or export. Oxygen concentrations were significantly more variable during the second cruise leg and did not exhibit clear diel patterns (Supplemental Fig. 9). Despite the intensification of photosynthesis and particle abundance at the DCM associated with the eddy, particle export at 150 m collected by sediment traps deployed across both legs during the entire duration of the sampling campaign were only slightly elevated compared to particle fluxes typically observed at Station ALOHA (Fig. 4), a finding consistent with grazing and particle remineralization in the upper ocean (<150 m).

Eddy weakening also manifested at the metatranscriptome level, with changes in the total gene expression, diversity, and function of microbial communities. Total transcript abundance (Kruskal-Wallis test, $p < 0.009$) and total RNA concentrations[35] were both higher during the 1st leg when the eddy was stronger. The resulting microbial metatranscriptome profiles exhibited differences between the two periods (Fig. 2, Supplemental Figs. 11, 12). Transcripts related to *Prochlorococcus*_B and eukaryotes were > 1.5× more abundant during the strong phase of the eddy (sampling Leg 1; Fig. 4). Key functional pathways also changed as the eddy varied in intensity, with an enrichment of KOs related to eukaryotic organisms that included a nitrate reductase (K10534) and transporter (K02575) when the eddy was stronger. In contrast, as the eddy weakened, transcripts related to the archaea *Nitrosopelagicus* (Thaumarchaeota), *Thalassarchaeum*

(MGII), and MGIIb-O1 all became more abundant (by 4×, 3.5×, and 1.7×, respectively) relative to the initial sampling period, including KOs specific to ammonia oxidation. Weakening of the eddy physical structure also appeared linked to a slight increase in the transcriptional activities of other distinct cyanobacterial taxa, including other LL clades (*Prochlorococcus*, *Prochlorococcus*_C, *Prochlorococcus*_D, AG-363-K07) and their photosynthetic genes (i.e., *pcb*, K08924; *psaI*, K02696). While putative iron transporters (K16091, K02012; heterotrophic bacteria and eukaryotes) and an iron-independent flavodoxin *isiB* (K03839; Cyanobacteria and *Nitrosopelagicus*) were more highly expressed when the eddy was stronger, the iron-requiring *petF* (K02639; Cyanobacteria) was more highly expressed as the eddy weakened (Supplemental Fig. 13, **Supplemental Information**).

## Discussion

By using AUVs capable of tracking and sampling the DCM in the NPSG at high temporal frequency, we find that microbial communities in the low-light waters of the photic zone undergo short-term (hourly scale) physiological and metabolic responses to diel changes in light, while simultaneously responding to intermediate-scale (days to weeks) changes in light and nutrients associated with mesoscale eddy physical forcing. The ability to autonomously sample a microbial population at the DCM in a Lagrangian manner allowed for an unprecedented view of gene expression and biogeochemistry over both hourly to weekly time scales. When coupled with long-term (monthly to decadal) measurements in the NPSG, this work highlights the time and space scales over which microbial communities respond to physical forcing in the open ocean.

While plankton metabolism and physiology have previously been shown to vary on diel scales in the well-lit upper ocean[9,12], this study documents daily-scale variation in microbial physiology and metabolism in the energy-limited waters of the DCM. In some cases, the diel-scale patterns observed in the DCM differed from those previously observed for well-lit waters: for example, we find chlorophyll fluorescence increases during the day, unlike at shallower depths where non-photochemical quenching reduces chlorophyll fluorescence during daylight hours[7,48,49]. However, we also see similarities in timing of gene expression in the DCM and shallower photic zone waters: for example, in both the DCM and well-lit upper ocean, phytoplankton photosynthesis, carbon fixation, and oxidative phosphorylation transcripts were all most abundant before and immediately after sunrise, with transcripts involved in cell division increasing later in the afternoon and evening[9,10,50]. While we document clear patterns of diel rhythmicity at the DCM, especially in the phytoplankton, when compared to the mixed layer, both the total number of unique diel transcripts (~10% vs <5% here) and their overall abundances within the dataset (up to ~50% vs <20% here) are lower at the DCM[32,51]. Our findings are consistent with the magnitude of diel gene expression decreasing with depth through the photic zone. The observation that less than 20% of all transcript expression in the DCM fluctuates on diel scales, along with a lack of clear oxygen periodicity as the eddy began to weaken, may explain why metabolomes from the lower photic zone during the second leg of this cruise did not show strong daily variability[52]. Overall, diel transcription by heterotrophs was low, although some transcripts appeared synchronously linked but not necessarily in ways that appeared tuned to variations in sunlight, a finding suggesting temporally complex metabolic regulation controlled by higher frequency environmental changes[8,53]. Our observations show that even in the poorly-lit waters of the DCM, daily temporal transcriptional dynamics persist and are manifest through observable fluctuations in biogeochemistry. Whether these observations extend to a typical DCM that is not impacted by a cyclone remains an open question. Such short-term variability in microbial physiology has important implications for the production and consumption of organic matter, nutrient cycling, and ultimately material export out of the upper ocean.

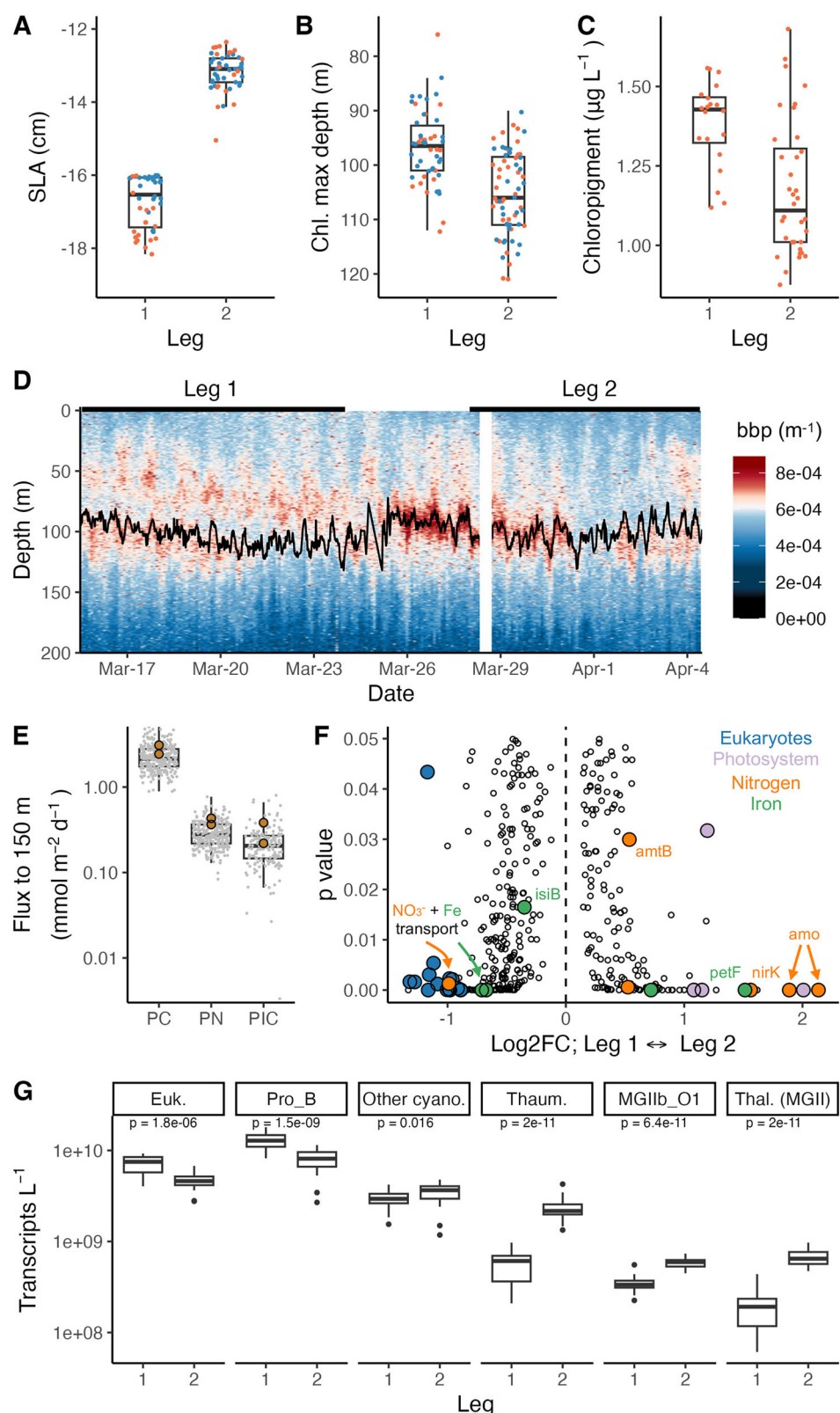

**Fig. 4 | Physiological and transcriptional changes are evident as an eddy weakens. A–C** Changes in sea level anomaly (SLA; **A**), the depth of the chlorophyll maximum layer (DCM; **B**), and chloropigment concentrations **C** across sampling legs. AUV data are shown in blue, and CTD rosette samples are in orange. **D** Backscatter during two successive Wirewalker deployments. The black line represents the DCM. Black lines at the top indicate approximate cruise leg time ranges. **E** The flux of particulate carbon (PC), particulate nitrogen (PN), and particulate inorganic carbon (PIC) measured by two sediment traps during the entire cruise (brown) relative to the HOT climatology (boxplots and gray dots). **F** KEGG orthologs identified as

differentially expressed during Leg 1 or Leg 2, with select orthologs colored and labeled. Log₂FC, log₂FoldChange. Transcript names include: *isiB*, flavodoxin; *amtB*, ammonium transporter; *petF*, flavodoxin; *ftn*, ferretin; *nirK*, nitrite reductase; *amo*, ammonia monooxygenase. **G** Transcript abundance (transcripts L⁻¹) per sample for specific taxonomic groups between sampling legs. Euk., eukaryotes; Pro_B, *Prochlorococcus*_B; Other cyano., other identifiable cyanobacterial genera; Thaum., Thaumarchaeota (or Thermoproteota); Thal. (MGII), Thalassarchaeum (MGII archaea). Boxplots throughout reflect the median, 25th, and 75th percentiles of data.

Superimposed on the daily rhythms in microbial activity were changes in community composition, metabolism, and physiology linked to variations in mesoscale eddy dynamics. Eddies are common in the NPSG, with more than 30% of observations at Station ALOHA influenced by the passage of these features[54–56]. Microorganisms in the DCM responded to changes in nutrients and light associated with isopycnal uplift. Concentrations of chlorophyll and photosynthetic picoeukaryote cell abundances were elevated compared to those typically observed at Station ALOHA, consistent with studies that have demonstrated that cyclonic eddies stimulate production by eukaryotic phytoplankton in the DCM[25,57,58]. While both *Prochlorococcus* and non-pigmented heterotrophic picoplankton abundances were lower than normal in the eddy DCM[55,59], both were enriched compared to concentrations typical for Station ALOHA along the isopycnal layer where the DCM resided, consistent with a prior response to isopycnal uplift. Elevated abundances and transcription by picoeukaryotes and LLI *Prochlorococcus* B, organisms which are normally found at and below the DCM, may point to the ability of these organisms to use nitrate and nitrite instead of ammonium[60–62], as well as the types of nitrogenous compounds in eddies influencing *Prochlorococcus* ecotype distributions[38,63]. While large photosynthetic taxa are known to respond to isopycnal uplift[64,65], our study was limited to plankton in the <5 µm fraction; hence, future work exploring transcriptional responses of larger eukaryotic phytoplankton to eddy isopycnal uplift would be valuable for improved understanding of how these organisms compete for resources in cyclonic eddies.

Eddies can perturb horizontal, vertical, and temporal distributions and activities of microbial communities. By using Lagrangian AUVs, we followed hourly to weekly scale microorganism transcriptional activities in the DCM, providing a mechanistic understanding of population-level responses to eddy weakening and isopycnal relaxation. Overall transcriptional activity was generally similar throughout the sampled period despite changes to both the physical and biological characteristics of the eddy (e.g., weakening of SLA, isopycnal relaxation, weakening and deepening of the DCM, and subsurface water mass horizontal displacement). Such findings are consistent with the hypothesis that phytoplankton biomass can remain elevated at the DCM weeks after intensification[25]. Although eddy isopycnal uplift may selectively stimulate microorganism growth rates and gene expression, such changes may not alter community composition per se because of tightly coupled removal processes. However, metatranscriptomic shifts from photosynthetic organisms toward ammonia-oxidizing archaea are consistent with a presumed biogeochemical transition from a community assimilating nitrate+nitrite towards one where reduced inorganic or organic nitrogen compounds become increasingly important as sources of energy. This transition coincides with the change from an eddy-induced phytoplankton bloom towards a decaying bloom state where remineralization of organic nitrogen drives ammonium production in the DCM, with shortened nitrogen remineralization length scales due to upward displacement of nitrifying microorganisms[66]. The enrichment of certain MGII archaea suggests microorganisms catalyzing protein degradation may respond during a remineralization-intensive phase of eddy weakening[39,67,68]. Cyclonic eddies have been shown to stimulate photosynthetic production without concomitant increases in organic carbon export flux[25,69,70], and our findings suggest organisms such as the MGII archaea and Thaumarchaeota (Thermoproteota) may contribute to rapid consumption of particulate material and reduced forms of nitrogen within the lower photic zone, prolonging the cycling of elements in the lower photic zone and temporally decoupling new production from export. While we attribute our findings to changes in eddy strength, the observed daily to weekly-scale variations in microbial dynamics may also be influenced by spatial differences across the eddy, mixing with external water, and divergent movement between surface and subsurface water masses. Remaining priorities for future work include

evaluating how microorganisms and their activities change spatially across an eddy and how they evolve throughout the stages (formation, maturation, decay) of an eddy's lifetime.

## Methods

A cyclonic eddy north of the Hawaiian Islands was surveyed during March–April 2018 on the Simons Collaboration on Ocean Processes and Ecology Hawaii Eddy Experiment (SCOPE HEE) cruise aboard the R/V *Falkor* (FK180310). The cruise consisted of two legs (Leg 1, March 10–24; Leg 2, March 28-April 4) and was followed by sampling of an adjacent anticyclone[52,64,71]. The eddy was identified using a satellite altimetry SLA product from the Copernicus Marine Environmental Monitoring Service[72]; http://marine.copernicus.eu/). The eddy was tracked using both the Mesoscale Eddy Trajectories Atlas (delayed time product version 3.2, cyclone n. 737324) distributed by AVISO+ (https://aviso.altimetry.fr/) and a simplified algorithm for determining eddy location, age, and strength (as inferred from SLA amplitude) as previously described[25]. SLA was corrected for interannual trend and seasonal cycles[55]. The coherency of the eddy was estimated using methods and data as described by Jones-Kellett & Follows[20,33].

The eddy was located ~200 km from the long-term research site Station ALOHA (22.75°N, 158.0°W). Station ALOHA has been sampled monthly (>350 cruises) since 1988 as part of the Hawaii Ocean Time-series (HOT) program[4,73]. Data collection and processing reported here were consistent with HOT methodologies and were compared against the long-term climatology at Station ALOHA.

### Vertical profiling of the hydrographic structure of the water column

Lagrangian sampling was performed using AUVs, drifters, and ship-based CTD casts. The eddy center was first identified using hydrographic surveys with an underway CTD (Teledyne, Falmouth, MA, USA). Two Surface Velocity Program (SVP) drifters (Pacific Gyre, Oceanside, CA, USA), with drogues centered at 15 m and 125 m, were deployed in the eddy center to follow both surface and subsurface (i.e., near the DCM) water masses. A surface-tethered, free-drifting Wirewalker profiler (Del Mar Oceanographic, San Diego, CA, USA) was deployed near the SVP drifters to obtain vertically resolved temperature, salinity, chlorophyll fluorescence, and backscatter to depths of 400 m. Two surface-tethered sediment trap arrays (one at 100-200 m near the eddy center, one at 150–500 m near the eddy edge), filled with seawater and sodium chloride (50 g L$^{-1}$) brine and formaldehyde as a preservative, were deployed for 15 days to collect vertically resolved sinking particulate material at 150 m as previously described[74]. The Lagrangian Wirewalker and sediment trap devices continued sampling the eddy while the ship was in port between legs; the Wirewalker was briefly recovered at the beginning of the second leg before being redeployed. The ship followed the SVP drifters and sampled in a quasi-Lagrangian manner: during the first leg, the ship followed the surface drifter, while during the second it followed the subsurface drifter. Maps were generated using MATLAB and the R package marmap v1.0.10[75,76].

Vertical profiles of hydrographic and biogeochemical properties were conducted ~ 3 times per day using a Sea-Bird SBE 911 plus CTD (Sea-Bird Scientific, Bellevue, WA, USA) affixed to a rosette bottle sampler. The rosette included a Wet Labs ECO-FLNTU fluorometer (Sea-Bird Scientific) and a Sea-Bird SBE 43 sensor for measurements of chlorophyll fluorescence and dissolved oxygen concentrations, respectively. CTD-based oxygen concentrations were standardized using Winkler titrations of discrete seawater samples collected from CTD casts. For our analyses, we focus on CTD casts and Wirewalker profiles conducted between March 16 (CTD cast 8) and April 3 (CTD cast 64), a time period that overlapped with AUV-collected metatranscriptomic data described below. For CTD casts, the depth of the DCM was defined as the depth of the highest chloropigment concentration

on a given cast. Wirewalker measurements, including backscatter at 650 nm, were plotted with ggplot2 v3.5.2[77] based on cubic spline interpolations calculated with the R package akima v0.6.3.4[78]. The depth of the DCM from the Wirewalker was estimated as the average depth where chlorophyll fluorescence exceeded 0.5 ug L$^{-1}$.

## Autonomous underwater vehicle sampling

Two custom-built *Tethys*-class long-range AUVs were used for tracking and sampling the DCM[34,35,79,80]. AUVs were deployed during two discrete time windows: Mar 17–22 (Leg 1) and Mar 28–Apr 2 (Leg 2). AUV *Opah* spiraled up and down the water column, measuring vertical variation in hydrographic properties. Meanwhile, AUV *Aku* autonomously tracked an isotherm corresponding to the depth of the peak of chlorophyll fluorescence (i.e., the DCM). This isotherm differed between the cruise sampling legs. The horizontal position of AUV *Aku* was driven by the current in a quasi-Lagrangian manner, performing 10 m radius circles while drifting. Both vehicles were fitted with a suite of sensors for measuring conductivity, temperature, pressure (depth), fluorescence, dissolved oxygen, and photosynthetically active radiation (PAR) as previously described[34]. Chlorophyll concentrations were calculated from fluorescence measurements following the manufacturer-provided formula. Oxygen concentrations from AUV *Aku* were standardized against shipboard-based oxygen determinations during the second leg when the AUVs and ship were in close proximity. Oxygen measurements from both assets were binned into 5 dbar increments and a linear correlation was used to standardize AUV *Aku* data. Environmental AUV data were interpolated and plotted as described above.

AUV *Aku* was equipped with a third-generation Environmental Sample Processor (ESP) capable of filtering and preserving samples in situ for metatranscriptomic analysis[9,10,81,82]. One sample was collected every 3 h at the DCM (61 total: 30 during Leg 1, 31 during Leg 2). Seawater (1.6 L) was sequentially filtered through a 5 μm polyvinylidene fluoride (PVF) membrane and onto a 0.22 μm PVF filter. Following filtration, the 0.22 μm filters containing the 0.22–5 μm cell size fraction were preserved with RNAlater. For each sample, in situ filtration took 65 min and processing took 12 min. Sampling times reported reflect the midpoint between the time filtering began and when samples were preserved in RNAlater. Upon recovery, samples were placed in microcentrifuge tubes and frozen at −80 °C.

## CTD rosette sampling

Picoplankton cell abundances and nutrient and chlorophyll *a* concentrations were determined from seawater samples collected using Niskin bottles affixed to the CTD sampling rosette. Samples were typically collected at fixed depth intervals (5, 25, 45, 75, 100, 125, 150, and 175 m), consistent with HOT sampling protocols, and on occasion from discrete depths around the DCM based on real-time vertical fluorescence profiles. Picoplankton cell abundances, including nonpigmented cells, *Prochlorococcus*, and picoeukaryotes (typically -1 μm), were determined based on scatter and fluorescence characteristics using an Influx Mariner flow cytometer (Cytopeia/BD, Franklin Lakes, NJ, USA) from samples collected on 9 individual CTD casts (8 from Leg 1, 1 from Leg 2). Samples for nutrient concentrations were collected from 10 CTD casts (8 from Leg 1, 2 from Leg 2) and frozen immediately. Concentrations of $NO_3^- + NO_2^-$ (N + N) and $PO_4^{3-}$ were determined by colorimetric analysis[83] using a SEAL Analytical AA3 (Mequon, WI, USA). Chlorophyll and phaeopigment concentrations were determined from seawater samples filtered onto 25 mm Whatman GF/F filters (Cytiva, Wilmington DE, USA) where pigments were extracted in acetone and quantified fluorometrically on a Turner Designs Model 10-AU fluorometer (Turner Designs, San Jose, CA, USA; 83). To contextualize the cell abundances and nutrient concentrations at the DCM (isopycnal range of AUVs across legs, -24.6–24.8 kg m$^{-3}$), we compared observations from this cruise to HOT program measurements typically found at the 24.7 kg m$^{-3}$ isopycnal at Station ALOHA.

Metatranscriptome samples were also collected from aboard the ship using the CTD rosette Niskin bottles. Samples were collected from the DCM daily at 06:00, 12:00, and 18:00 h HST. A total of 36 samples were collected across both legs. Seawater (2 L) was filtered (80 mL min$^{-1}$) through a 5 μm PVF membrane and onto a 0.22 μm PVF filter using a peristaltic pump at 80 mL min$^{-1}$. Following filtration, the 0.22 μm filters containing the 0.22–5 μm cell size fraction were submerged in RNAlater and frozen at −80 °C. Sampling times reported reflect the beginning of the CTD cast.

## RNA extraction and sequencing

After transport to the shore-based laboratory, filters were thawed on ice, briefly centrifuged, and RNAlater removed. Filters were treated with 300 μL of Ambion denaturing lysis buffer (Thermo Fisher Scientific, Waltham, MA, USA). To allow for quantitative estimates of RNA concentrations, an External RNA Controls Consortium (ERCC) spike in RNA standard (2 μL of a 1:100 dilution) was added to each sample at a concentration expected to reflect 1% of total sample reads. Samples were then amended with 898 μL of nuclease-free water. RNA was purified from 650 μL of each sample using a Chemagic Magnetic Separation Module instrument (PerkinElmer, Waltham, MA, USA) with the Tissue RNA kit (CMG-1212, PerkinElmer). After DNase treatment, magnetic bead purification, and the removal of rRNA by Ribozero treatment (Illumina, San Diego, CA, USA), RNA was eluted with 50 μL of RNase-free water and frozen at −80 °C. RNA quality was assessed using a Fragment Analyzer with High Sensitivity RNA reagents (Agilent, CA, USA) and quantified using Ribogreen (Invitrogen, Thermo Fisher Scientific). The ScriptSeq v2 RNA-Seq kit (Illumina #SSV21124, San Diego, CA, USA) was used to create cDNA and sequencing libraries with barcodes following the manufacturer's guidelines with an EpMotion 5075 TMX (Eppendorf, Hamburg, Germany). Libraries were normalized to a final DNA concentration of 4 nmol L$^{-1}$ and pooled in equal volumes. A phiX control (Illumina) was added to an estimated final contribution of 5% of the total estimated sequence depth. DNA was sequenced as 150 bp paired-end reads on an Illumina NextSeq 500 using a V2 high-output 300-cycle reagent kit.

## Metatranscriptome analyses

Paired-end reads were quality screened prior to assembly. First, Illumina adapters were removed using the bbduk script from BBMap v39.01[84]. A second pass using bbduk removed phiX sequences, low-quality bases, ribosomal RNA, and sequences with spurious GC content. Error correction was performed using BFC[85], and additional low-quality bases were removed using Trimmomatic v0.39[86]. Unpaired reads were removed using SeqTK 1.3-r106. Samples were assembled individually using RNA-SPAdes v3.15.5[87]. Predicted genes were identified using Prodigal v2.6.3[88] and clustered with the existing ALOHA 2.0 gene catalog[89] at 95% similarity using CD-HIT v4.8.1[90]. Cleaned transcripts were mapped to the combined gene catalog using last[91]. Gene counts were normalized relative to the abundances of spike-in ERCC standards of known concentration and to the volume of water filtered for each sample, producing quantitative counts of transcripts L$^{-1}$.

Gene cluster representatives were annotated against multiple databases using sequence homology. For both functional and phylogenetic annotation, transcripts were annotated against the EggNOG 5.0 database[92] using the eggNOG-mapper v2.1.10[93] at an e-value cutoff of 0.001. KEGG annotations obtained using EggNOG were supplemented by mapping transcripts against an in-house KEGG database using last and KO categories linked to their function and pathway using KEGGREST v1.44.1[94,95]. Further taxonomic assignments were made using the Genome Taxonomy Database for Bacteria and Archaea (GTDB v214[96]) and EUKulele[97]. Transcripts were first classified taxonomically using GTDB, with the last with a percent identity threshold

of 90%. If sequences fell below this threshold, they were then classified using EggNOG broad RefSeq annotations with an e-value threshold of e[-40]. EUKelele was then used to taxonomically identify transcripts identified as eukaryotic using RefSeq and those that remained unidentified. Sequences were annotated as eukaryotic if they hit both the MarRef-MMETSP[98,99] and EukZoo databases[100]. If sequences remained unidentified, they were classified as unknown.

We identified transcripts that differed over the diel cycle, between cruise legs, and between AUV versus shipboard CTD casts. Transcripts showing diel periodicity across a 24 h cycle were determined using nonparametric rhythm detection with the R package RAIN v1.38[101]. Each cruise leg and sampling methodology (AUV or CTD) was treated independently. First, the dataset was subset to transcripts present in at least 10 samples across the entire dataset ( > ~10% of samples). Linear trends were removed from each transcript using the R package pracma v2.4.6[102]. Diel transcripts were identified using RAIN with the following parameters: period = 24, deltat = 3 (or 6 for CTD), method = independent. *P*-values were adjusted for false discovery rates using Benjamini-Hochberg correction[103]. Transcripts were considered as showing evidence of diel periodicity at a corrected *p*-value threshold of <0.05 for AUV samples and 0.1 for CTD samples. The time of maximum expression for each transcript and lines of best fit to show oscillating patterns were determined using loess local regression in R. Modules of co-expressed transcripts were determined using the package WGCNA v1.73 on the 100,000 most expressed transcripts[104]. A soft threshold of 5, the smallest value that would achieve an $R^2$ value > 0.9 for fit to a scale-free topology, was chosen using the "pickSoftThreshold" command. Transcriptional modules were identified using the "blockwiseModules" command with a minimum module size of 100 and an eigengene correlation threshold of <0.4. Modules with eigengene dissimilarities <0.25 were merged, and modules composed of transcripts that were enriched in less than two samples were excluded from analysis. Differentially abundant transcripts, KEGG Ortholog (KO) categories, and taxonomic groups between cruise legs and sampling methodologies were identified using DeSEQ2 v1.44[105] with significance defined at an adjusted *p*-value < 0.05. To identify differences in KO categories between cruise legs, KO categories were summed within each sample prior to analysis. Similar results were obtained when controlling for time using an additive model (~time + leg). To identify transcripts showing differential expression between cruise legs and sampling methodologies, DeSEQ2 comparisons were performed on transcripts without rarefaction, but were subset to the top 300,000 most expressed transcripts to remove low abundance sequences[106]. Comparisons of overall expression of taxonomic groups between cruise legs were also performed using Kruskal-Wallis tests. Further discussion of these comparisons can be found in the **Supplemental Information**.

## Data availability

Raw sequence data are available in the National Center for Biotechnology Information (NCBI) Short Read Archive (SRA) archive under BioProject accession number PRJNA962143 (Supplementary Data 1). Cruise data, including CTD and AUV deployment data, cell abundances, and nutrient concentrations, are available at the Simons Collaborative Marine Atlas Project titled Falkor_2018. HOT program data are available at https://hahana.soest.hawaii.edu/hot/. Representative code and parameters can be found in the Supplemental Material (Supplementary Code 1).

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

## Acknowledgments

We thank those involved in the success of the R/V *Falkor* cruise FK180310 and the Hawaii Ocean Time-series, including James Birch, Gabe Foreman, Thomas O'Reilly, Steve Poulos, Ben-Yair Raanan, Hans Ramm, Brent Roman, Paul Den Uyl, Elisha Wood-Charlson, Blake Watkins, the crew of the R/V *Falkor*, and all others who helped develop and deploy the AUVs. We also acknowledge Alexandra Jones-Kellett for helpful discussions on eddy coherency and spatial variability and Lance Fujieki for assistance with oxygen calibrations. We are grateful for funding from the Simons Foundation as part of the Simons Collaboration

on Ocean Processes and Ecology (grants 721221 to M.J.C., 721223 to E.F.D., 721252 and 721264 to D.M.K., and 329108 to D.M.K. and E.F.D.), the Simons Foundation International (Life Sciences grants 00010767 to M.J.C., 00010668 to E.F.D., 00009452 to D.M.K., and 00009481 to B.B.), the David and Lucile Packard Foundation (Monterey Bay Aquarium Research Institute), and the Schmidt Ocean Institute.

## Author contributions

E.F.D., D.M.K., and C.S. conceived the study. B.B., B.W.H., B.K., R.M. III, C.M.P., A.E.R., J.P.R., S.T.W., and Y.Z. performed fieldwork, sample collection, and laboratory work. L.M.P. and J.M.E. performed sequence and data analyses. L.M.P., M.J.C., and E.F.D. wrote the manuscript with input from the other authors. R.M. III is deceased, and this manuscript was written and published posthumously in his honor.

## Competing interests

The authors declare no competing interests.
