## [Peer Review file · Nature Communications]

Diel and eddy driven changes in microbial gene expression and biogeochemistry in the oceanic chlorophyll maximum

Corresponding Author: Dr Logan Peoples

Version 0:

Reviewer comments:

Reviewer #1

(Remarks to the Author)

Summary:

This study by Peoples et al., applied an impressive array of cutting-edge technology (AUVs equipped with ESPs, Wirewalkers, etc) to perform an in situ LaGrangian metatranscriptomic and biogeochemical investigation of diel periodicity in the low-light DCM of a cyclonic eddy situated near HOTs. This study has many strengths, including the use of various temporal scales (hourly, weekly decadal), the pairing of metatranscriptomic data with biogeochemical analyses, and the contextualization of the data via its posit with the long-term HOTs dataset. Many interesting findings were uncovered in this study, including the prominence of diel periodicity expression at a depth where light has rapidly attenuated. The finding that light continues to play a primary role in community expression and physiology at such a depth is interesting in itself, and the subsequent exploration of these findings (including the prominence of the *Pelagibacter bacteriorhodopsin* expression) merit discussion. Overall, I enjoyed reading this manuscript and believe it to be a great fit for Nature Comms. However, I have a few questions and concerns which I have noted below which may warrant investigation prior to publication. While there are small discrepancies in grammar and method reporting, my primary concern relates to the diel analysis. I did not see clear evidence of detrending prior to the RAIN analysis – which could skew results. In addition, comparable diel studies in the photic zone of the Sargasso Sea and Northern Pacific report ~8-11% of genes (across domains) exhibiting diel periodicity, and this study reports nearly double that (22%). While this could be an interesting finding meriting further discussion / comparison in the manuscript, it also may seem a bit counter-intuitive when thinking of the attenuation of light. I suggest the authors elaborate on this finding to ensure methodological proceedings (detrending, statistical cut-offs, etc.) are not the primary cause.

Overall, I commend the authors on a great study and look forward to seeing this work published subsequently after the revision stage.

Best regards,
Brittany Zepernick

Overarching comments:

-I believe the manuscript would benefit from a bit of wordsmithing and edits to grammar, sentence structure and diction throughout. In my specific comments I have highlighted a few examples of sentences that don't make sense or convolute the message.

-In addition, there are discrepancies in the methods which hinder reproducibility of the study. This is especially pertinent in the metatranscriptomic analysis sections. I recommend revising these methods to ensure the study is both clear to the reader and reproducible. In addition, providing the scripts used to process the metatranscriptomes and subsequent RAIN analysis via github or Zenodo is an optional (additional) step towards reproducibility.

-My foremost concern: I have questions regarding the RAIN analysis and detection of diel gene expression. The data did not

appear to be detrended prior to the RAIN analysis – which may have skewed the number of diel genes detected. This leads to my second question, this study reports ~22% of genes exhibited diel periodicity across all domains, which seems high to me considering other studies which applied the same analysis to pelagic, marine environments routinely report ~8-11% of genes exhibiting periodicity across domains (this is elaborated on in my specific comments below and the studies are cited). I would appreciate if the authors could clarify if the data was detrended prior to RAIN analysis and also compare their results from the marine DCM to other studies relating to the photic zone of marine systems.

Specific comments:

-Title: I suggest changing the title to a concise, definitive statement which summarizes the core finding. As it reads now., it is a bit vague and the takeaway is unclear. I have provided a few examples of recent Nature Communications articles (published October 2025) that do a good job of this (in my opinion):

Ex. “Structure-function relationship of the GH168 fucanase reveals an unusual enzyme recognition mechanism for sulfated polysaccharide”

Ex. “Autophagy acts as a brake on obesity-related fibrosis by controlling purine nucleoside signaling”

Ex. “Backtracking metabolic dynamics in single cells predicts bacterial replication in human macrophages”

-Line 97: “Production of organic carbon by plankton fuels tightly connected oceanic food webs whose collective metabolism consumes most of this primary production before it leaves the sunlit layer of the upper ocean (i.e., photic zone)”. The sentence directly before also uses the verb “fuels” which is redundant, and the current sentence is a bit confusing in the underlined portion. I recommend simplifying the sentence structure to a more definitive statement here and using a variety of verbs.

-Line 103: I suggest limiting the references to three maximum per statement as is common practice. Listing > five references (as done in this specific instance) is a bit excessive.

-Line 108: “...spanning orders of magnitude in time and space scales...”. Should this not be “spanning orders of magnitude across temporal and spatial scales”?

-Line 155: This is a run-on sentence which becomes confusing. I recommend simplifying this to concise, definitive statements. The key take-aways are there yet diluted by the sentence structure.

-Line. 177: The coordinates appear to be presented in the Degree, Decimal, Minutes format – please convert to Decimal Degrees format as this is common publishing practice.

Methods: Throughout the methods, equipment is listed yet the manufacturer and the location of headquarters is not provided. This is common practice for equipment and key consumables to facilitate reproducibility. I recommend making this change throughout. As an example: Qubit RNA HS Assay Kit (Invitrogen, Waltham, MA, USA).

Methods: Throughout the methods, programs are used yet the version is not listed. This again hinders reproducibility. For example, Line 209: what version of the R package “akima” was used? What version of “marmap” in Line 198?

-Line 233: I assume the ~30 RNA samples collected via the AUV ESP for each leg have no biological replication? Was one filter collected every three hours? I suggest explicitly stating replication to ensure the sample matrix is clear to the reader. In addition, I am a bit concerned with the length of filtration time (65 minutes filtration and 12 minutes of processing). I can appreciate the revolutionary nature of the AUV ESP and its ability to collect in situ RNA samples (super cool!). However, many would be hesitant to sequence a sample that took 65 minutes to filter on a research vessel / in the lab. After this time, I would be concerned the cells would be transcriptionally responding to the filtration and accumulation on the filter itself... rather than the environmental conditions they were collected from. Could the authors elaborate on this step and explain any efforts made to avoid false interpretations?

-Line 245: “Samples were collected at fixed depth intervals (5, 25, 45, 75, 100, 120,150 and 175) and on occasion from discrete depths around the DCM”. Could the authors elaborate on why those fixed depth intervals were selected and if the occasional DCM samples were based on specific observations? Alternatively, were the occasional DCM samples purely opportunistic?

-Line 287 and throughout: This is a bit picky, but typically a transcriptome refers to the expression profile of one taxa / culture and hence it is used often when describing laboratory studies. Environmental communities spanning all domains etc., are typically referred to as metatranscriptomes / metatranscriptomics. I suggest updating the terminology for clarity.

-Line 289: “Transcriptomic analyses were quality screened using BBMap and Trimmomatic...”. Were the reads also filtered and / or trimmed in these steps as is standard (especially when using Trimmomatic)? Were the default settings used? If so, I recommend stating this. If not, please list the command / filter specifications. Ex: Low quality reads were removed and adapters subsequently trimmed via Trimmomatic (leading = 20, trailing = 20, sliding window = 4:15, minimum length = 36) (v.0.39) (Bolger et al., 2014). In addition, which BBMAP script was used to quality assess the transcripts? Typically, I am accustomed to this program for interleaving reads, mapping reads, etc. and the use of it as described here is unfamiliar.

-Line 290: Were efforts made to remove residual rRNA or contaminants in silico? While a Ribozero treatment was applied

post-extraction, these treatments are best amended by subsequent in silico removal of residual rRNA.

-Line 291: It appears the libraries were individually assembled as opposed to a single, concatenated coassembly? Would the authors clarify which method was used and please state so explicitly?

-Line 299: Which version of the EggNOG database was used and was a specified e-value used to screen the hits? If so, please state this. In addition, eggNOG-mapper results typically print out annotations for PFAMs, KEGG, COGs, GO, and BRITe in addition to the descriptions / hits to the eggNOG reference database. I am also curious why eggNOG-mapper and KEGG database were both used? This seems otherwise redundant unless the authors have found a discrepancy between eggNOG hits to KEGG vs. directly annotating with KEGG?

-As an aside, I commend the authors on their extremely robust taxonomic screening! The programs used maximize taxonomic annotations for all domains – especially the often-overlooked eukaryotes. I am curious what the rough % was of unknowns after this rigorous screening – it would be a great statistic to report here for those looking to increase their annotations and update lab pipelines!

-Line 319: Were the transcripts detrended before the RAIN analysis was performed? Detrending is calculating the linear regression with respect to time of the time series and subsequently subtracting it. This increases the power of rhythmicity. In particular, I note this study found ~22% of the total transcripts exhibited diel trends and this is double the values that have been reported prior in the literature. For example, Muratore et al., 2025– reported ~11% of genes across marine domains exhibited diel expression in the photic zone of the Sargasso Sea. A similar number was also reported in the North Pacific Subtropical Gyre (~9% of genes exhibited diel periodicity across domains) in Muratore et al., 2022. This ~doubling in diel genes detected in this study could be due to variances in detrending, statistical thresholds etc., or it could suggest diel patterns are more strongly detected in the low-light DCM compared to the upper photic zone (which the aforementioned studies reported on). The latter would be very interesting / exciting! Though, this would be a bit counter-intuitive considering the inferred role of light in diel periodicity. I have provided the full references of the aforementioned below and would appreciate if the authors elaborated on their recovery of diel genes in the DCM compared to other diel periodicity studies conducted in the Sargasso Sea, the Northern Pacific, etc. Contextualizing results in this manner would be of benefit to the study (in my opinion) and also help satisfy concerns about detrending etc.

REFS:

Muratore, D., Gilbert, N. E., LeClerc, G. R., Wilhelm, S. W., & Weitz, J. S. (2025). Diel partitioning in microbial phosphorus acquisition in the Sargasso Sea. *Proceedings of the National Academy of Sciences*, 122(11), e2410268122.

Muratore, D., Boysen, A. K., Harke, M. J., Becker, K. W., Casey, J. R., Coesel, S. N., ... & Weitz, J. S. (2022). Complex marine microbial communities partition metabolism of scarce resources over the diel cycle. *Nature Ecology & Evolution*, 6(2), 218-229.

-I am curious, given the multiple spatiotemporal scales in this study, how DESEQ2 was applied to this dataset? Was the additive or interactive model used to deduce DE genes and what statistical threshold was used to discern differential expression (padj and LG2FC?)

-Line 427: The discussion of rhodopsins is particularly interesting!! Though given the large array of rhodopsin classifications, I suggest the authors elaborate on the type of rhodopsins that were recovered if they subsequently attribute them to phototrophic energy generation. Given there are xanthorhodopsins, proton-pumping rhodopsins, Cl⁻ pumping rhodopsins, Na⁺ pumping rhodopsins, xenorhodopsins, sensory rhodopsins, bacteriorhodopsin, etc. – and each has a distinct predicted protein structure, function and localization – this may be noteworthy to report (and potentially shift the interpretation). The two KEGG KO's in Supplemental Figure 8 are indeed bacteriorhodopsin and a sensory rhodopsin according to the KEGG manuscript reference – yet these KEGG KO's are based on manuscripts from the 1990's and much progress has been made in the rhodopsin front since then...though we still have a long ways to go and many proton-pumping rhodopsins are not annotated correctly via functional programs such as eggNOG. A quick confirmation of the protein structure via Protter or the presence of a retinal Schiff base would prove very compelling for the interpretation of phototrophic energy generation. The work by Oded Beja and Laura Gomez-Consarnua on rhodopsins may be of particular help with this.

Reviewer #2

(Remarks to the Author)

Comments to the authors are attached.

Reviewer #1 (Remarks to the Author):

Summary:

This study by Peoples et al., applied an impressive array of cutting-edge technology (AUVs equipped with ESPs, Wirewalkers, etc) to perform an in situ LaGrangian metatranscriptomic and biogeochemical investigation of diel periodicity in the low-light DCM of a cyclonic eddy situated near HOTs. This study has many strengths, including the use of various temporal scales (hourly, weekly decadal), the pairing of metatranscriptomic data with biogeochemical analyses, and the contextualization of the data via its posit with the long-term HOTs dataset. Many interesting findings were uncovered in this study, including the prominence of diel periodicity expression at a depth where light has rapidly attenuated. The finding that light continues to play a primary role in community expression and physiology at such a depth is interesting in itself, and the subsequent exploration of these findings (including the prominence of the Pelagibacter bacteriorhodopsin expression) merit discussion. Overall, I enjoyed reading this manuscript and believe it to be a great fit for Nature Comms. However, I have a few questions and concerns which I have noted below which may warrant investigation prior to publication. While there are small discrepancies in grammar and method reporting, my primary concern relates to the diel analysis. I did not see clear evidence of detrending prior to the RAIN analysis – which could skew results. In addition, comparable diel studies in the photic zone of the Sargasso Sea and Northern Pacific report ~8-11% of genes (across domains) exhibiting diel periodicity, and this study reports nearly double that (22%). While this could be an interesting finding meriting further discussion / comparison in the manuscript, it also may seem a bit counter-intuitive when thinking of the attenuation of light. I suggest the authors elaborate on this finding to ensure methodological proceedings (detrending, statistical cut-offs, etc.) are not the primary cause.

Overall, I commend the authors on a great study and look forward to seeing this work published subsequently after the revision stage.

We thank the reviewer for their overall positive comments and helpful suggestions on our manuscript. We have responded to each point, including those described above, in detail below.

Overarching comments:

-I believe the manuscript would benefit from a bit of wordsmithing and edits to grammar, sentence structure and diction throughout. In my specific comments I have highlighted a few examples of sentences that don't make sense or convolute the message.

Thank you. We have updated the text where specified by the reviewer.

-In addition, there are discrepancies in the methods which hinder reproducibility of the study. This is especially pertinent in the metatranscriptomic analysis sections. I recommend revising these methods to ensure the study is both clear to the reader and reproducible. In addition, providing the scripts used to process the metatranscriptomes and subsequent RAIN analysis via github or Zenodo is an optional (additional) step towards

reproducibility.

To address this comment, we have now updated the methods section to more clearly describe the analyses performed. We have also now included a supplemental text file that provides representative parameters used to run the bioinformatic analyses in the manuscript, along with the code necessary to reproduce Figures 1-4. This includes flags and code for metatranscriptomic analysis, RAIN analysis, and differential abundance analysis using DESeq2.

-My foremost concern: I have questions regarding the RAIN analysis and detection of diel gene expression. The data did not appear to be detrended prior to the RAIN analysis – which may have skewed the number of diel genes detected. This leads to my second question, this study reports ~22% of genes exhibited diel periodicity across all domains, which seems high to me considering other studies which applied the same analysis to pelagic, marine environments routinely report ~8-11% of genes exhibiting periodicity across domains (this is elaborated on in my specific comments below and the studies are cited). I would appreciate if the authors could clarify if the data was detrended prior to RAIN analysis and also compare their results from the marine DCM to other studies relating to the photic zone of marine systems.

The data had not previously been detrended. Following the recommendation of the reviewer, we have now detrended the data to remove linear changes in expression that may have occurred over the course of each leg using the R package *pracma* prior to performing periodicity analyses. This has now been updated in the methods. Our results and our interpretation have not changed (see below). The methods now read as follows:

“Transcripts showing diel periodicity across a 24 hr cycle were determined using nonparametric rhythm detection with the R package RAIN v1.38 (Thaben & Westermark, 2014). Each cruise leg and sampling methodology (AUV or CTD) were treated independently. First, the dataset was subset to transcripts present in at least 10 samples across the entire dataset (> ~10% of samples). Linear trends were removed from each transcript using the R package *pracma* v2.4.6 (Borchers, 2025). Diel transcripts were identified using RAIN with the following parameters: period = 24, *deltat* = 3 (or 6 for CTD), method = independent. P values were adjusted for false discovery rates using Benjamini-Hochberg correction (Benjamini & Hochberg, 1995). Transcripts were considered as showing evidence of diel periodicity at a corrected p value threshold of < 0.05 for AUV samples and 0.1 for CTD samples.”

We would like to emphasize that in the manuscript, we are reporting three distinct things: 1) the total number of unique diel transcripts, 2) the diversity of the unique diel transcripts, and 3) the combined, total *expression* (abundance) of diel transcripts summed across the entire dataset relative to the total transcript expression across the entire dataset. Our updated analyses identified 13,735 unique transcripts that exhibited evidence of diel periodicity across the AUV samples. When their expression values are summed across the entire dataset, these diel transcripts reflect 19.4% of the total expressed transcript abundances, similar to the 22% we previously stated. If we increase the stringency to a corrected p value of 0.01, 6275 unique transcripts oscillate. These transcripts still reflect 15% of the total summed transcript expression across the entire dataset.

Thank you to the reviewer for providing two references (in a separate comment below) describing diel periodicity in the North Pacific Subtropical Gyre and Sargasso Sea. In these references, the authors state that 8.9-11.4% of genes showed evidence of diel periodicity. We interpret this as the percentage of unique transcripts which show diel periodicity. On a smaller dataset, Vislova et al. report that as much as 17-36% of unique transcripts show evidence of diel rhythmicity at 25 m. In our dataset we identified 13,735 unique transcripts: these transcripts reflect 0.3% of all unique transcripts (> 4 million), 3.8% of the transcripts screened (those present in > 10 samples, ~358,000), or 4.2% of the top 100,000 most expressed transcripts (4159 unique transcripts). Regardless of the denominator used, our results show that a smaller percentage of unique transcripts oscillate at the DCM when compared to those reported in the mixed layer, consistent with what might be expected.

Our statement that ~20% of total transcript expression exhibits diel rhythmicity takes into account the *expression* level of these unique transcripts: when summing up the total abundances of all (quantitative) RNA expressed across our time series, what percentage do diel transcripts represent? To our knowledge, the references provided by the reviewer do not report this statistic. At 25 m, Vislova et al. report that diel-expressed transcripts can represent ~20% (March) to more than 50% (June) of all expressed transcript abundances, again higher than that reported here. Therefore, when considering both 1) the number of unique transcripts which show diel expression and 2) their level of expression, a significant fraction of expressed RNA molecules at the DCM oscillate along a diel cycle, but at apparently lower abundances than in the mixed layer.

To clarify our findings, we have now added the following text to the Results:

“Coinciding with these daily changes, we identified 13,735 transcripts that exhibited significant diel oscillations, with 3,449 of these showing periodicity during both sampling legs (**Figure 3, Supplemental Table 2**). Rhythmic transcripts represented 0.3% of all unique transcripts (> 4 million), 3.8% of the transcripts screened (those present in > 10 samples, ~358,000), and 4.2% the top 100,000 most expressed transcripts (4159 unique transcripts). When considering the expression level of transcripts by summing the quantitative abundances of diel-expressed transcripts relative to the expression of all transcripts across the entire dataset, diel-expressed transcripts comprised 19.4% of total expressed transcript abundance, highlighting the importance of diel variability in transcription even in the poorly lit waters of the DCM.”

We have also added the following text to the Discussion:

“While we document clear patterns of diel rhythmicity at the DCM, especially in the phytoplankton, when compared to the mixed layer both the total number of unique diel transcripts (~10% vs < 5% here) and their overall abundances within the dataset (up to ~50% vs < 20% here) are lower at the DCM (Vislova *et al.*, 2019; Muratore *et al.*, 2025). Our findings are consistent with the magnitude of diel gene expression decreasing with depth through the photic zone. The observation that ‘only’ ~20% of all transcript expression in the DCM fluctuates on diel scales may explain why metabolomes from the lower photic zone during the second leg of this cruise did not show strong daily variability (Kumler *et al.*, 2024). Overall, diel transcription by heterotrophs was low, although some transcripts appeared synchronously linked but not

necessarily in ways that appeared tuned to variations in sunlight, a finding suggesting temporally complex metabolic regulation controlled by higher frequency environmental changes (e.g., Ottesen *et al.*, 2013; Aylward *et al.*, 2015). Our observations show that even in the poorly-lit waters of the DCM, daily temporal transcriptional dynamics persist and are manifest through observable fluctuations in biogeochemistry. Such short-term variability in microbial physiology has important implications for the production and consumption of organic matter, nutrient cycling, and ultimately material export out of the upper ocean.”

Vislova et al., 2019 <https://doi.org/10.3389/fmicb.2019.02191>

Specific comments:

-Title: I suggest changing the title to a concise, definitive statement which summarizes the core finding. As it reads now., it is a bit vague and the takeaway is unclear. I have provided a few examples of recent Nature Communications articles (published October 2025) that do a good job of this (in my opinion):

Ex. “Structure-function relationship of the GH168 fucanase reveals an unusual enzyme recognition mechanism for sulfated polysaccharide”

Ex. “Autophagy acts as a brake on obesity-related fibrosis by controlling purine nucleoside signaling”

Ex. “Backtracking metabolic dynamics in single cells predicts bacterial replication in human macrophages”

We have updated the title. The title now reads: “Diel and eddy driven changes in microbial gene expression and biogeochemistry in the oceanic chlorophyll maximum”.

-Line 97: “Production of organic carbon by plankton fuels tightly connected oceanic food webs whose collective metabolism consumes most of this primary production before it leaves the sunlit layer of the upper ocean (i.e., photic zone)”. The sentence directly before also uses the verb “fuels” which is redundant, and the current sentence is a bit confusing in the underlined portion. I recommend simplifying the sentence structure to a more definitive statement here and using a variety of verbs.

Thank you. The sentence has been updated, including replacing the verb “fuels” with “supports”. The sentence now reads: “Organic carbon produced by phytoplankton supports tightly connected oceanic food webs that consume most primary production in the sunlit layer of the upper ocean (i.e., the photic zone).”

-Line 103: I suggest limiting the references to three maximum per statement as is common practice. Listing > five references (as done in this specific instance) is a bit excessive.

We have removed references such that only three are now listed in this sentence. In order to further streamline the text as suggested, we have limited the number of references to three in each clause throughout the manuscript.

-Line 108: “...spanning orders of magnitude in time and space scales...”. Should this not be “spanning orders of magnitude across temporal and spatial scales”?

Thank you. The sentence has been updated and split into two sentences. They now read: “Plankton are closely connected to an array of heterogeneous physical features that define the ocean habitat (Stommell, 1963). Microorganism growth and physiology can respond rapidly to variations in energetic and nutritional resources triggered by these physical features.”

-Line 155: This is a run-on sentence which becomes confusing. I recommend simplifying this to concise, definitive statements. The key take-aways are there yet diluted by the sentence structure.

This comment was referring to the final sentence of the Introduction. The sentence has now been simplified to emphasize our findings. It now reads: “We find that microbial populations respond to changes in both diurnal variations in light and mesoscale eddy strength, emphasizing how different time and space scales influence plankton physiology and biogeochemistry in the ocean.”

-Line. 177: The coordinates appear to be presented in the Degree, Decimal, Minutes format – please convert to Decimal Degrees format as this is common publishing practice.

The Latitude and Longitude are now presented as decimal degrees.

Methods: Throughout the methods, equipment is listed yet the manufacturer and the location of headquarters is not provided. This is common practice for equipment and key consumables to facilitate reproducibility. I recommend making this change throughout. As an example: Qubit RNA HS Assay Kit (Invitrogen, Waltham, MA, USA).

The text has been updated to contain all manufacturer names, including headquarter locations, at first mention.

Methods: Throughout the methods, programs are used yet the version is not listed. This again hinders reproducibility. For example, Line 209: what version of the R package “akima” was used? What version of “marmap” in Line 198?

Version numbers have now been included for packages, including marmap, akima, ggplot2, RAIN, WGCNA, and DeSEQ2.

-Line 233: I assume the ~30 RNA samples collected via the AUV ESP for each leg have no biological replication? Was one filter collected every three hours? I suggest explicitly stating replication to ensure the sample matrix is clear to the reader. In addition, I am a bit concerned with the length of filtration time (65 minutes filtration and 12 minutes of processing). I can appreciate the revolutionary nature of the AUV ESP and its ability to collect in situ RNA samples (super cool!). However, many would be hesitant to sequence a sample that took 65 minutes to filter on a research vessel / in the lab. After this time, I

would be concerned the cells would be transcriptionally responding to the filtration and accumulation on the filter itself.... rather than the environmental conditions they were collected from. Could the authors elaborate on this step and explain any efforts made to avoid false interpretations?

We have updated the text to emphasize that one sample was collected every three hours. The text now reads: “One sample was collected every 3 hrs at the DCM (61 total: 30 during Leg 1, 31 during Leg 2).”

We acknowledge that filtration time may impact community composition. Sampling using traditional CTD-based Niskin bottles shipboard also has drawbacks – for example, by exposing microbial communities to non-native conditions (Feike et al., 2012) or by missing particles (e.g. Suter et al., 2017). While the communities here do show differences based on sampling methodology, transcript expression overall looks similar, we do not observe expression patterns that would be reflective of a filtration response (for example, surface attachment), and previous analyses using the AUVs suggest that AUV and traditional shipboard sampling methods produce similar transcriptional profiles (Ottesen et al., 2011). Furthermore, it is hard to establish whether differences between AUV and CTD rosette-collected samples are derived from sampling methodology or reflect true biological variability, for example due to differences in sampling location within the eddy or due to slightly different depths of collection.

To acknowledge the possibility that AUV sampling may in some way disturb these communities, we have modified the Supplemental Material to acknowledge this: “Transcriptomic differences across samples collected by AUV versus CTD rosette could be due to both sampling methodologies and small-scale spatial heterogeneity within the eddy. From the time samples are collected, brought shipboard, and processed, traditional CTD sampling takes longer (2 hrs +) than AUV sampling (~1.5 hrs). CTD rosette-collected communities are exposed to atypical conditions when brought shipboard, causing perturbations in light and temperature that may vary cell physiology. Hence, the filtration time during AUV sampling could induce transcriptional changes, although changes likely also occur during shipboard sampling. The AUV samples were preserved with RNA later immediately following filtration but were retained at temperatures of the chlorophyll maximum for multiple days. This delay in temperature preservation may impact RNA integrity, although both previous estimates (Ottesen *et al.*, 2011) and those as part of this study (Zhang *et al.*, 2021) suggest no loss of RNA integrity.”

Ottesen et al., 2011. <https://doi.org/10.1038/ismej.2011.70>

Feike et al., 2012. <https://doi.org/10.1038/ismej.2011.94>

Suter et al., 2017 <https://doi.org/10.1002/lno.10447>

-Line 245: “Samples were collected at fixed depth intervals (5, 25, 45, 75, 100, 120,150 and 175) and on occasion from discrete depths around the DCM”. Could the authors elaborate on why those fixed depth intervals were selected and if the occasional DCM samples were based on specific observations? Alternatively, were the occasional DCM samples purely opportunistic?

Sampling was performed at fixed depths intervals consistent with HOT sampling protocols, allowing us to compare these measurements with the Station ALOHA North Pacific Subtropical Gyre monthly climatology. Because this cruise was focused in part on the deep chlorophyll maximum, targeted sampling occurred at depths around the DCM; however, this was not performed on every cast, and therefore we state that these events occurred “on occasion”.

To clarify, the text has been updated and now reads: “Samples were typically collected at fixed depth intervals (5, 25, 45, 75, 100, 125, 150, and 175 m), consistent with HOT sampling protocols, and on occasion from discrete depths around the DCM based on real-time vertical fluorescence profiles.”

-Line 287 and throughout: This is a bit picky, but typically a transcriptome refers to the expression profile of one taxa / culture and hence it is used often when describing laboratory studies. Environmental communities spanning all domains etc., are typically referred to as metatranscriptomes / metatranscriptomics. I suggest updating the terminology for clarity.

Transcriptome/transcriptomics has been replaced with metatranscriptome/metatranscriptomics throughout the text.

-Line 289: “Transcriptomic analyses were quality screened using BMAP and Trimmomatic...”. Were the reads also filtered and / or trimmed in these steps as is standard (especially when using Trimmomatic)? Were the default settings used? If so, I recommend stating this. If not, please list the command / filter specifications. Ex: Low quality reads were removed and adapters subsequently trimmed via Trimmomatic (leading = 20, trailing = 20, sliding window = 4:15, minimum length = 36) (v.0.39) (Bolger et al., 2014). In addition, which BMAP script was used to quality assess the transcripts? Typically, I am accustomed to this program for interleaving reads, mapping reads, etc. and the use of it as described here is unfamiliar.

Thank you for these comments. The bbdduk script from BMAP was used to remove adapters, phiX sequences, ribosomal RNA, low quality bases, and sequences with spurious GC content. Trimmomatic was run using the parameters LEADING:10 TRAILING:10 MINLEN:50.

Two changes have been made. First, we have now included a supplemental file that contains detailed parameters and flags used for commands for processing the metatranscriptome data. This is now stated in the “Data Availability” section. Second, the methods have been updated to state the filtering methods in more detail. They now read:

“Paired-end reads were quality screened prior to assembly. First, Illumina adapters were removed using the bbdduk script from BMAP v39.01 (Bushnell et al., 2014). A second pass using bbdduk removed phiX sequences, low-quality bases, ribosomal RNA, and sequences with spurious GC content. Error correction was performed using BFC (Li et al., 2015) and additional low-quality bases were removed using Trimmomatic v0.39 (Bolger et al., 2014). Unpaired reads were removed using SeqTK 1.3-r106. Samples were assembled individually using RNA-SPAdes v3.15.5 (Bushmanova et al., 2019). Predicted genes were identified using Prodigal v2.6.3 (Hyatt

et al., 2010) and clustered with the existing ALOHA 2.0 gene catalog (Luo *et al.*, 2020) at 95% similarity using CD-HIT v4.8.1 (Fu *et al.*, 2012). Cleaned transcripts were mapped to the combined gene catalog using last (Kiełbasa *et al.*, 2011).”

-Line 290: Were efforts made to remove residual rRNA or contaminants in silico? While a Ribozero treatment was applied post-extraction, these treatments are best amended by subsequent in silico removal of residual rRNA.

Yes, residual rRNA was removed using bbduk, which identifies rRNA based on kmers.

-Line 291: It appears the libraries were individually assembled as opposed to a single, concatenated coassembly? Would the authors clarify which method was used and please state so explicitly?

Yes, the samples were assembled separately with SPAdes (3.15.5) with the parameters: "--rna -k 55". These parameters have been added to a supplemental file and the text now reads: “Samples were assembled individually using RNA-SPAdes v3.15.5 (Bushmanova *et al.*, 2019).”

-Line 299: Which version of the EggNOG database was used and was a specified e-value used to screen the hits? If so, please state this. In addition, eggNOG-mapper results typically print out annotations for PFAMs, KEGG, COGs, GO, and BRITe in addition to the descriptions / hits to the eggnog reference database. I am also curious why eggNOG-mapper and KEGG database were both used? This seems otherwise redundant unless the authors have found a discrepancy between eggNOG hits to KEGG vs. directly annotating with KEGG?

We used eggnog mapper 2 (version 2.1.10) with default options against the EggNOG database 5.0 with an e-value cutoff of 0.001. To confirm these annotations, we supplemented them by mapping against an in-house KEGG database using last. To clarify, the text now reads: “For both functional and phylogenetic annotation, transcripts were annotated against the EggNOG 5.0 database (Huerta-Cepas *et al.*, 2019) using the eggNOG-mapper v2.1.10 (Cantalapiedra *et al.*, 2021) at an e-value cutoff of 0.001. KEGG annotations obtained using EggNOG were supplemented by mapping transcripts against an in-house KEGG database using last and KO categories linked to their function and pathway using KEGGREST v1.44.1 (Kanehisa *et al.*, 2016; Tenenbaum, 2024).”

-As an aside, I commend the authors on their extremely robust taxonomic screening! The programs used maximize taxonomic annotations for all domains – especially the often-overlooked eukaryotes. I am curious what the rough % was of unknowns after this rigorous screening – it would be a great statistic to report here for those looking to increase their annotations and update lab pipelines!

The “unknown” category reflects on average 39% of the metatranscriptome abundances. The following sentence is stated in the Results: “Unclassified transcripts averaged 39% of the total transcript expression per sample.”

-Line 319: Were the transcripts detrended before the RAIN analysis was performed? Detrending is calculating the linear regression with respect to time of the time series and subsequently subtracting it. This increases the power of rhythmicity. In particular, I note this study found ~22% of the total transcripts exhibited diel trends and this is double the values that have been reported prior in the literature. For example, Muratore et al., 2025– reported ~11% of genes across marine domains exhibited diel expression in the photic zone of the Sargasso Sea. A similar number was also reported in the North Pacific Subtropical Gyre (~9% of genes exhibited diel periodicity across domains) in Muratore et al., 2022. This ~doubling in diel genes detected in this study could be due to variances in detrending, statistical thresholds etc., or it could suggest diel patterns are more strongly detected in the low-light DCM compared to the upper photic zone (which the aforementioned studies reported on). The latter would be very interesting / exciting! Though, this would be a bit counter-intuitive considering the inferred role of light in diel periodicity. I have provided the full references of the aforementioned below and would appreciate if the authors elaborated on their recovery of diel genes in the DCM compared to other diel periodicity studies conducted in the Sargasso Sea, the Northern Pacific, etc. Contextualizing results in this manner would be of benefit to the study (in my opinion) and also help satisfy concerns about detrending etc.

REFS:

Muratore, D., Gilbert, N. E., LeCclair, G. R., Wilhelm, S. W., & Weitz, J. S. (2025). Diel partitioning in microbial phosphorus acquisition in the Sargasso Sea. *Proceedings of the National Academy of Sciences*, 122(11), e2410268122.

Muratore, D., Boysen, A. K., Harke, M. J., Becker, K. W., Casey, J. R., Coesel, S. N., ... & Weitz, J. S. (2022). Complex marine microbial communities partition metabolism of scarce resources over the diel cycle. *Nature Ecology & Evolution*, 6(2), 218-229.

Please see our response above to this reviewer about detrending and percentages of unique transcripts vs percent of total expression.

-I am curious, given the multiple spatiotemporal scales in this study, how DESEQ2 was applied to this dataset? Was the additive or interactive model used to deduce DE genes and what statistical threshold was used to discern differential expression (padj and LG2FC?)

DESeq2 was applied to this dataset using a design that took into account only the cruise leg and not the time of collection. An adjusted p value (padj) of < 0.05 was used to identify differential expression (with rare KOs excluded). Two plots below are provided to highlight that when controlling for time, similar results are obtained. The first plot shows KO categories enriched in Leg 1 vs Leg 2 using an additive model with DESeq (design = ~day_night+leg) where samples were grouped into day (6 AM to 6 PM) and night (6 PM to 6 AM) categories. The results from this analysis are very similar to that reported in Figure 4F. To further confirm our findings, the second plot below shows the absolute abundances of six representative KO categories which are highlighted (colored) in Figure 4F. This figure indicates that the KO categories identified using our DESeq analysis do show significant (Kruskal-Wallis test), quantitative differences between

legs. KO categories shown: K02012, afuA, iron (III) transport system protein; K02639, petF, ferredoxin; K02575, narK/nasA, nitrate/nitrite transporter; K10944, amoA/pmoA, ammonia/methane monooxygenase subunit A; K10408, dynein chain motor protein; K12856, mRNA processing factor.

The methods have been updated to more clearly reflect our methods. They now read: “Differentially abundant transcripts, KEGG Ortholog (KO) categories, and taxonomic groups between cruise legs and sampling methodologies were identified using DeSEQ2 v1.44 (Love *et al.*, 2014) with significance defined at an adjusted p value < 0.05. To identify differences in KO categories between cruise legs, KO categories were summed within each sample prior to

analysis. Similar results were obtained when controlling for time using an additive model (~time+leg). To identify transcripts showing differential expression between cruise legs and sampling methodologies, DeSEQ2 comparisons were performed on transcripts without rarefaction but were subset to the top 300,000 most expressed transcripts to remove low abundance sequences (McMurdie & Holmes, 2014).”

-Line 427: The discussion of rhodopsins is particularly interesting!! Though given the large array of rhodopsin classifications, I suggest the authors elaborate on the type of rhodopsins that were recovered if they subsequently attribute them to phototrophic energy generation. Given there are xanthorhodopsins, proton-pumping rhodopsins, Cl⁻ pumping rhodopsins, Na⁺ pumping rhodopsins, xenorhodopsins, sensory rhodopsins, bacteriorhodopsin, etc. – and each has a distinct predicted protein structure, function and localization – this may be noteworthy to report (and potentially shift the interpretation). The two KEGG KO’s in Supplemental Figure 8 are indeed bacteriorhodopsin and a sensory rhodopsin according to the KEGG manuscript reference – yet these KEGG KO’s are based on manuscripts from the 1990’s and much progress has been made in the rhodopsin front since then...though we still have a long ways to go and many proton-pumping rhodopsins are not annotated correctly via functional programs such as eggNOG. A quick confirmation of the protein structure via Protter or the presence of a retinal Schiff base would prove very compelling for the interpretation of phototrophic energy generation. The work by Oded Beja and Laura Gomez-Consarnua on rhodopsins may be of particular help with this.

Thank you for your comment. This is in reference to a statement in the main text that *Pelagibacter* may be capable of “phototrophic energy generation” (see below for referenced text). We recognize the high diversity of rhodopsins, especially within mixed marine communities like those described here, and that KO categories may not be the best way to classify or identify them. Identifying the specific types of rhodopsins in great detail is outside the scope of this manuscript, and we note in the text that these are “putative rhodopsins”. However, to evaluate our statement about their importance in *Pelagibacter* specifically, we have now performed analyses identifying residues that have been implicated in ion pumping and rhodopsin spectral tuning as described in Olson *et al.*, 2018. We focused our analysis on the top 50 most highly expressed transcripts identified as rhodopsins by KO category. The results of this analysis can be found below in panel C of an updated Supplemental Figure 8. The vast majority of these sequences were identified as related to the Pelagibacterales (SAR11 or *Pelagibacter*), with smaller contributions by the Flavobacteriales and SAR86, consistent with the reported broader distribution (Panel A). Motif analysis of specific amino acids involved in ion pumping found that all sequences had the amino acids DTE (aspartate, threonine, and glutamine), indicating these proteins act as proton-pumping rhodopsins rather than sodium or chloride pumps (Yoshizwa *et al.*, 2016; Olson *et al.*, 2018). Analysis of specific amino acids involved in spectral tuning found that all sequences had Q and M at the position described in Olson *et al.* and Yoshizwa *et al.*: these amino acids allow for blue and green light absorption, respectively. Therefore, our findings are consistent with previous analyses at Station ALOHA that have found 1) high abundances of SAR11-related proteorhodopsins and 2) the dominance of proton-pumping, blue-light tuned rhodopsins (DTE-Q) with minor contributions (but the highest of any depth) of green-light DTE-M rhodopsins at the DCM.

The original text states: “Based on transcriptional patterns in the DCM, *Pelagibacter* appears to couple transport of these compounds to phototrophic energy generation: up to 50% of total putative rhodopsin gene expression was linked to the Pelagibacterales (Supplemental Figure 8).”

Supplemental Figure 8. The relative expression per AUV sample of rhodopsin (A) and tonB-dependent receptor (B) KEGG orthologs by taxon. C) Amino acid motif analysis of the top 50 most highly expressed rhodopsin transcripts in the dataset. Amino acid motif DTE-Q reflects proton-pumping, blue-light tuned rhodopsins, while motif DTE-M reflects proton-pumping, green-light tuned rhodopsins (see Olson *et al.*, 2018).

Olson DK, Yoshizawa S, Boeuf D, Iwasaki W, DeLong EF. 2018. Proteorhodopsin variability and distribution in the North Pacific Subtropical Gyre. *ISME J* 12: 1047-1060.

<https://doi.org/10.1038/s41396-018-0074-4>

Yoshizawa, Y. Kumaga, H. Kim, Y. Ogura, T. Hayashi, W. Iwasaki, E.F. DeLong and K. Kogure. 2014. Functional characterization of flavobacteria rhodopsins reveals a unique class of light-driven chloride pump in bacteria. *Proc Natl Acad Sci USA* 111: 6732-6737.

<https://doi.org/10.1073/pnas.1403051111>

Reviewer 2

Summary of Article:

The article “Diel and eddy-induced temporal variability in microbial gene expression and biogeochemistry at the chlorophyll maximum” aims to address 3 research questions: (1) How does eddy isopycnal displacement impact microbial communities in the DCM? (2) To what extent do microorganisms in the DCM show evidence of diel periodicity in gene expression? And (3) How does microbial population structure and physiology change as the eddy decays? To achieve these goals, the authors performed extensive Lagrangian sampling of one cyclonic eddy located north of the Hawaiian Islands and contextualized their findings using data from Station ALOHA. The authors found (1) that isopycnal uplift increased the abundance of heterotrophic picophytoplankton, *Prochlorococcus*, and photosynthetic picoeukaryotes relative to the 24.7 kg m⁻³ isopycnal at Station ALOHA, (2) that 22% of the total expressed transcripts showed diel periodicity at the DCM, and (3) that the initiation of eddy decay resulted in a shift in microbial composition and key functional pathways.

The manuscript is of high interest not only because it reveals novel patterns in microbial community activity in the DCM at a variety of temporal resolutions but also because it reveals how changes in physical forcings impact microbial community composition and function using a Lagrangian sampling approach. Overall, the manuscript is well written and provides thorough analyses. Only minor revisions are required prior to publication. The revisions primarily center on providing additional methodological details, clarifying some of the results, improving figures to aid in reader interpretation, and moving important details from the supplemental information to the main text. These revisions are outlined in detail below.

Thank you for your comments and help improving our manuscript.

Specific Comments:

Abstract:

1. Line 65: The eddy appears to be at the very beginning stages of decay. Please be specific in word choice here, “As the eddy began to decay”.

The text has been updated as recommended. The word “decayed” has been replaced with “began to weaken.”

Introduction:

2. Line 155: The study does not follow the entire process of eddy decay. I recommend changing “eddy decays” to “eddy weakens” or “eddy begins to decay”.

The text has been updated and now reads: “(3) How does microbial population structure and physiology change as an eddy weakens?”

Furthermore, all instances of “decay” have been replaced with “weakening” throughout the text.

Methods:

3. Line 209: Please state the interpolation method that is implemented by “akima”.

A cubic spline interpolation was implemented using the akima package. The text has been updated and now reads: “Wirewalker measurements, including backscatter at 650 nm, were plotted with ggplot2 v3.5.2 (Wickham, 2011) based on cubic spline interpolations calculated with the R package akima v0.6.3.4 (Akima *et al.*, 2016).

4. Lines 309 – 317: The filtration methodology used for CTD and AUV samples (0.22 – 5 um) is not adequate for retention of free-living viruses. With this study design, only attached and intracellular viruses were retained. Since viruses are not a major component of this manuscript, I would recommend removing this altogether. However, if the authors wish to include it, these methodological limitations must be clearly addressed within the main text. Additionally, it should be clearly stated in the results that any recovered viral transcripts are from attached or intracellular viruses and do not represent the entire viral community.

The reviewer is correct: these analyses would only capture intracellular viruses as they are replicating. As recommended by the reviewer, we have removed discussion of viruses from the manuscript and figures.

5. Line 321: Please describe how RAIN statistically determines periodicity.

RAIN uses nonparametric rhythm detection and makes no assumptions about the shape of the periodicity of transcripts. The text has now been updated to read: “Transcripts showing diel periodicity across a 24 hr cycle were determined using nonparametric rhythm detection with the R package RAIN v1.38 (Thaben & Westermark, 2014).”

6. Line 335: State what threshold was used to define low abundance sequences.

This question is describing the DESeq2 analysis performed. The dataset contains a large number of sequences at low abundances which can provide spurious or uninformative results. Prior to these analyses, the dataset was subset to the top 300,000 transcripts, representing 80% of the total transcript expression.

The text has been updated to now read: “To identify transcripts showing differential expression between cruise legs and sampling methodologies, DeSEQ2 comparisons were performed on transcripts without rarefaction but were subset to the top 300,000 most expressed transcripts to remove low abundance sequences (McMurdie & Holmes, 2014).”

7. Line 346: Please include your R and MATLAB code as part of the publicly available data.

We have now included a supplemental file that includes many of the representative flags and the code used to prepare Figures 1-4.

Results:

8. Lines 351 – 354: Either move these lines to the Introduction or remove them, since they are not results from this study.

This comment was referring to the first few lines of the results which stated that eddies are common in the NPSG. These sentences have been removed from the results section.

9. Line 357 and throughout: Eddy “strength” was not directly measured in this study. While higher eddy kinetic energy can lead to higher amplitudes, amplitude is not a direct measure of eddy strength. Please choose a different word throughout.

SSH is commonly used to estimate eddy strength. However, we acknowledge that eddy strength and intensity may not always correlate with sea surface height amplitude and sea level anomaly. The data collected in our study suggest that eddy strength was correlated with SLA: increases in the depth of isopycnal density layers, along with decreases in chlorophyll concentrations, occurred during a period of more positive SLA. To address the reviewer’s comment, the text in both the methods and results have been updated to emphasize that eddy strength is being inferred from sea level anomaly.

The text on the specified line now reads: “Hindcasting suggests eddy genesis occurred in early February 2018 with the eddy moving southward while intensifying to its maximum strength (as inferred by amplitude; SLA of -18 cm) in March (Supplemental Figure 1).”

10. Line 375: Phosphate concentrations are not shown in Figure 1. Please add in a reference to Supplemental Figure 5B to direct the reader to the correct figure.

We have modified the text so that the reader is now directed to both Figure 1 and Supplemental Figure 5.

11. Line 379: Supplemental Figure 5D also shows that there is a higher abundance of photosynthetic picoeukaryotes than is typical for the 24.7 kg m⁻³ isopycnal at Station ALOHA. Please add this result to the main text as it is relevant to addressing your first research question.

The text has now been updated to include this observation. It now reads: “Elevated chlorophyll concentrations and abundances of photosynthetic picoeukaryotes, in comparison to typical conditions at both the DCM and the 24.7 kg m⁻³ isopycnal, indicate that the upward displacement of nutrient-enriched water stimulated primary production (Figure 1G-J, Supplemental Figure 5D).

12. Line 381: Similar to Comment #10; heterotrophic picoplankton abundances are not shown in Figure 1. Please add in a reference to Supplemental Figure 5F.

The text has now been updated to send interested readers to both Figure 1K and Supplemental Figure 5E-G.

13. Line 387: The phrasing is a bit confusing because Figure 2 appears to only show the AUV samples while Supplemental Figure 6 shows the AUV and CTD samples. Please be very clear throughout the results about which figures and analyses use AUV + CTD

samples versus only the AUV samples.

Throughout the manuscript, we largely focus on results from the AUV samples as they were collected at higher temporal resolution (every 3 hrs). Results from the CTD rosette-collected samples are also included but largely discussed in the supplemental material. To clarify this, we have updated Figure 2 to only include AUV samples and added a sentence to the beginning of the results. The text now reads: “Quantitative metatranscriptomic sequencing was performed on 97 samples, 61 from AUV *Aku* and 36 collected using the CTD rosette. We mostly focus on results from AUV samples that were collected every ~3 hrs (see Supplemental Information).”

14. Line 392: Please be a little more specific in word choice and change “most active” to “most active Cyanobacteria”.

The text has been updated to include “Cyanobacteria”.

15. Lines 400 – 401: Please state the average relative transcript expression for the eukaryotic genera listed in these lines.

The text has been updated and now reads: “Transcripts identified as belonging to the Eukaryota represented ~6% of each sample, with the most abundant group being *Pelagomonas* (2%) and smaller contributions from *Bathycoccus* (0.4%), *Chrysochromulina* (0.4%), *Ostreococcus* (0.3%), MAST-4A (0.2%), *Prymnesium* (0.2%).”

16. Lines 406 – 407: Please state the average expression per transcript for the genera listed in these lines.

These sentences have been removed as they were largely tangential and included some discussion rather than strictly results.

17. Lines 417 – 420: Please state which genera and KOs you’re referring to.

To address this, a new supplemental table has been added to the manuscript. The text also now reads: “The most highly expressed KO categories across the entire dataset reflected the importance of photosynthesis and carbon fixation, cell division and replication, and the uptake of both organic and inorganic compounds, especially nitrogen in the form of ammonium, proteins, and amino acids (Supplemental Table 2).”

18. Lines 432 – 435: These KOs are not represented in Figure 2C, please add in the correct figure reference or state in the text that this is “data not shown”.

This comment references a list of transcripts and the KO categories expressed by the MGII archaea. There is no figure to reference for this list; therefore, we have added “data not shown” as requested.

19. Line 436: Please state the KOs for ammonia oxidation and add in a reference to Figure 2 after “(Thaumarchaeota)”. This will help to distinguish between the data that is represented in Figure 2 versus Supplemental Figure 8.

Both the KO for the ammonia monooxygenase subunit A and a reference to Figure 2 have been added to the text.

20. Line 437: tonB-dependent receptor in AG-430-B22 is not shown in either Figure 2 or Supplemental Figure 8. Please either revise your figures to include it or remove it from the text.

The expression of transcripts encoding for the tonB-dependent receptor KO K02014 in AG-430-B22 is shown in both Figure 2 and Supplemental Figure 8. In Figure S8, it is represented by the order Rhizobiales_B.

21. Line 444: Change “fluorescence” to “chlorophyll concentrations”.

“Fluorescence” has been replaced with “fluorescence-derived chlorophyll concentrations”.

22. Line 445: Please define the hours that are “morning”, “afternoon”, “evening”, and “night” before using these terms to describe the results.

Times (approximate hours) have now been added throughout this paragraph to provide more detail on the timing of gene expression.

23. Line 445 – Line 449: Please explain why only data from leg 1 of the cruise was analyzed and represented in Figure 3A – 3C.

When showing the timing of chlorophyll and oxygen concentrations relative to PAR, data from Leg 1 was used for two reasons. First, the PAR sensor on AUV *Aku* was not working during the second leg. Second, oxygen concentrations were significantly more variable during Leg 2 and did not exhibit clear diel patterns, although highest concentrations were still in the late afternoon and evening. We hypothesize that this is because the eddy weakened and transitioned towards a net heterotrophic community that consumed particles, leading to higher rates of consumption and habitat oxygen heterogeneity. Data from Leg 2 has now been included as Supplementary Figure 9, reproduced below.

Text has also been added to the results describing oxygen variability. It reads: “Oxygen concentrations were significantly more variable during the second cruise leg and did not exhibit clear diel patterns (Supplemental Figure 9).”

24. Line 465 – 467: As above for Comment #4.

The sentence on viruses has been removed.

25. Line 503: Please describe how the communities differ between leg 1 and leg 2. For example, reductions in the abundance of Cyanobacteria.

The sentences following the line referenced here describe the results requested: there are reductions in transcripts by the Cyanobacteria and eukaryotes and increases in transcript abundances related to members of the archaea.

26. Line 503: It is unclear to me how Supplemental Figures 12 and 13 show differences in microbial community composition.

Supplemental Figure 12 shows groups of transcripts that are more highly expressed during the second sampling leg, indicating differences associated with eddy weakening. Clustering samples as shown in Supplemental Figure 13B shows that samples collected on the same leg are more similar to one another than those collected on different legs.

Two changes have been made to clarify these findings. First, we have replaced the term “microbial community composition” in the text with “microbial metatranscriptome profiles” to more clearly state what was measured. The text now reads: “The resulting microbial metatranscriptome profiles exhibited differences between the two periods (**Figure 2, Supplemental Figure 11, Supplemental Figure 12**).”. Second, the text in Supplemental Figure

12 has been updated to highlight the purpose of the figure. It now reads: “Groups of co-expressed transcripts that show higher expression during one sampling leg relative to the other but do not oscillate across a diel cycle”.

27. Line 508: Please add in a reference to Figure 4, and please make it clear in the text that *Nitrosopelagicus* is represented as Thaumarchaeota in Figure 4G.

A reference to Figure 4 has been added to the preceding sentence, along with a clause stating that *Nitrosopelagicus* is within the Thaumarchaeota.

Discussion:

28. Line 525: Please change “decay” to “weakening”.

Decay has been replaced with weakening.

29. Line 586: Please also include distance from eddy center as an important changing factor.

Text has now been added to the results describing changes in subsurface water mass sampling location relative to the center of the eddy. It now reads: “The horizontal position of subsurface water masses sampled within the eddy also changed over time, with the AUVs sampling further from the center of the eddy during the second leg (**Supplemental Figure 4**). These physical and spatial changes, in part associated with eddy weakening, had clear influences on biology (**Figure 4**).”

We have also added text to the conclusion to highlight the importance of these spatial differences. It now reads: “While we attribute our findings to changes in eddy strength, the observed daily to weekly-scale variations in microbial dynamics may also be influenced by spatial differences across the eddy, mixing with external water, and divergent movement between surface and subsurface water masses.”

Figures:

30. Figure 1A: Please state in the figure caption what the arrows on the map are representing.

The legend for Figure 1 now contains the following explanation: “Arrows reflect the direction and strength of absolute geostrophic currents.”

31. Figure 1B: Please state why a section of data is missing or omitted from this plot.

Chlorophyll data is not available during March 18/19 because AUV *Opah* temporarily surfaced. The figure legend now contains the following sentence: “AUV-derived chlorophyll concentrations were unavailable when AUV *Opah* periodically surfaced (white regions in panel B).”

32. Figure 2A: Please state in the figure caption what “Time” represents and how this corresponds to sampling location in the eddy.

For clarity, the X axis sample labels had been removed and all samples had been ordered based on their sampling date and time, as reflected by “Time”. We have now updated the X axis to include the day of collection, with the first sample collected on each day labeled. The figure legend has been changed to now read: “Sample names are removed for clarity but are ordered by time of collection from left to right with the first sample collected on each day labeled. Leg 1 samples were obtained when the eddy was stronger, while Leg 2 samples were obtained as the eddy began to weaken.”

33. Figure 4D: Please annotate the panels as leg 1 and leg 2.

Black lines have been added to the top of Figure 4D to reflect approximate times of each leg.

34. Figure 4E: Please add the individual measurements of PC, PN, and PIC flux at Station ALOHA as points.

Individual measurements have now been added to Figure 4E.

35. Supplemental Figure 1:

- a. Please add panel labels to the figure.**
- b. Please state in the figure caption what the black circle is in the left panel.**
- c. Please add to the right panels, something that denotes when leg 1 and leg 2 of the cruise occurred. This will help the reader better interpret how the collected samples align with the process of eddy decay.**

Panel labels, a description of the black circle, and colors to denote the two sampling legs have now been added. The figure legend now reads: “In **A**, the dotted line reflects the track of the center of the eddy and the black circle reflects the location of long-term research location Station ALOHA. In **B+C**, the orange and blue lines reflect Leg 1 and Leg 2, respectively.”

36. Supplemental Figure 2:

- a. Please add panel labels.**
- b. Please add red dots to denote where transcriptomic samples were collected (like in Figures 1B and 1C)**

Panel labels and grey dots have been added to denote where metatranscriptomic samples were collected.

37. Supplemental Figure 3:

- a. Please make the shapes representing CTD and AUV sample locations larger.**
- b. Please add a boundary to the green shapes to make them more distinguishable.**

This comment refers to a map showing sampling locations. Shapes have been made larger and now include black boundaries to make them more distinguishable.

38. Supplemental Figure 4:

a. Please add panel labels.

b. Please define what SLA_{corr} minimum is in the figure caption.

Panel labels (A and B) have now been added and a description of SLA_{corr} minimum has now been added.

39. Supplemental Figure 5: Please increase the font of the R² and p-values.

The font size of the R² and p values has been increased.

40. Supplemental Figure 6: Please state in the figure caption how sample collection time corresponds with sample position in the eddy.

The figure has been updated to now include the date of sample collection along the X axis to clarify “Time”. A sentence has also been added to the legend to state how the sample collection date corresponds with eddy SLA. The figure legend now reads: “Sample names are removed for clarity but are ordered by time of collection from left to right with the first sample collected on each day labeled. Leg 1 samples were obtained when the eddy was stronger, while Leg 2 samples were obtained as the eddy began to weaken.”

41. Supplemental Figure 9: It is not possible to see the diel expression of individual transcripts in this plot. Please choose different line colors or line types to help distinguish patterns of individual transcripts.

This figure showed viral transcripts. This figure has now been removed (please see previous comments to this reviewer).

42. Supplemental Figures 10 and 11: It is unclear what the difference between these figures is.

This comment refers to two figures that showed groups of co-expressed transcripts. Many of these groups of transcripts show evidence of diel expression. There was no “difference” between the two figures: in total they showed seven different groups of transcripts, and were separated for ease of visualization purposes, ordered by group size. In response to this comment, the two figures have now been combined into one large figure.

43. Supplemental Figures 10, 11, and 12: Please state what the two different panels within the left panel represent. I suspect that they represent the two different legs of the cruise, but this needs to be clarified.

The reviewer is correct: they reflect the two sampling legs of the cruise. The two panels are now labeled with the appropriate cruise leg across all listed figures.

Supplemental Information:

44. Line 11; Lines 17 - 20: These lines of text highlight that there is an increase in sampling distance from the eddy center during leg 2 of the cruise and that this increase in distance influences both the water column properties and the biological patterns. This is a major result that affects the interpretation of the study. Please include the effect of distance from eddy center on the observed patterns in the main text.

We have now updated the text in the final paragraph of the Discussion describing changes in eddy strength to acknowledge that spatial differences likely play an important role in our interpretation of microbial activity within eddies. The text now reads: “While we attribute our findings to changes in eddy strength, the observed daily to weekly-scale variations in microbial dynamics may also be influenced by spatial differences across the eddy, mixing with external water, and divergent movement between surface and subsurface water masses. Remaining priorities for future work include evaluating how microorganisms and their activities change spatially across an eddy and how they evolve throughout the stages (formation, maturation, decay) of an eddy’s lifetime.”

45. Line 108: Add a comma after Cyanobacteria.

A comma has been added following Cyanobacteria.

46. The supplemental information contains a lot of pertinent information for interpreting the results of this study. In the main text, please refer the reader more often to the relevant corresponding supplemental information.

We have referred the reader to relevant corresponding supplemental figures and information more frequently as requested by this reviewer in their previous comments.

ROUND 1 REVIEWER 2 ATTACHMENT:

Summary of Article:

The article “Diel and eddy-induced temporal variability in microbial gene expression and biogeochemistry at the chlorophyll maximum” aims to address 3 research questions: (1) How does eddy isopycnal displacement impact microbial communities in the DCM? (2) To what extent do microorganisms in the DCM show evidence of diel periodicity in gene expression? and (3) How does microbial population structure and physiology change as the eddy decays? To achieve these goals, the authors performed extensive Lagrangian sampling of one cyclonic eddy located north of the Hawaiian Islands and contextualized their findings using data from Station ALOHA. The authors found (1) that isopycnal uplift increased the abundance of heterotrophic picophytoplankton, *Prochlorococcus*, and photosynthetic picoeukaryotes relative to the 24.7 kg m⁻³ isopycnal at Station ALOHA, (2) that 22% of the total expressed transcripts showed diel periodicity at the DCM, and (3) that the initiation of eddy decay resulted in a shift in microbial composition and key functional pathways.

The manuscript is of high interest not only because it reveals novel patterns in microbial community activity in the DCM at a variety of temporal resolutions but also because it reveals how changes in physical forcings impact microbial community composition and function using a Lagrangian sampling approach. Overall, the manuscript is well written and provides thorough analyses. Only minor revisions are required prior to publication. The revisions primarily center on providing additional methodological details, clarifying some of the results, improving figures to aid in reader interpretation, and moving important details from the supplemental information to the main text. These revisions are outlined in detail below.

Specific Comments:

Abstract:

1. Line 65: The eddy appears to be at the very beginning stages of decay. Please be specific in word choice here, “As the eddy began to decay”.

Introduction:

2. Line 155: The study does not follow the entire process of eddy decay. I recommend changing “eddy decays” to “eddy weakens” or “eddy begins to decay”.

Methods:

3. Line 209: Please state the interpolation method that is implemented by “akima”.
4. Lines 309 – 317: The filtration methodology used for CTD and AUV samples (0.22 – 5 µm) is not adequate for retention of free-living viruses. With this study design, only attached and intracellular viruses were retained. Since viruses are not a major component of this manuscript, I would recommend removing this altogether. However, if the authors wish to include it, these methodological limitations must be clearly addressed within the main text. Additionally, it should be clearly stated in the results that any recovered viral transcripts are from attached or intracellular viruses and do not represent the entire viral community.
5. Line 321: Please describe how RAIN statistically determines periodicity.
6. Line 335: State what threshold was used to define low abundance sequences.
7. Line 346: Please include your R and MATLAB code as part of the publicly available data.

Results:

8. Lines 351 – 354: Either move these lines to the Introduction or remove them, since they are not results from this study.
9. Line 357 and throughout: Eddy “strength” was not directly measured in this study. While higher eddy kinetic energy can lead to higher amplitudes, amplitude is not a direct measure of eddy strength. Please choose a different word throughout.
10. Line 375: Phosphate concentrations are not shown in Figure 1. Please add in a reference to Supplemental Figure 5B to direct the reader to the correct figure.
11. Line 379: Supplemental Figure 5D also shows that there is a higher abundance of photosynthetic picoeukaryotes than is typical for the 24.7 kg m⁻³ isopycnal at Station ALOHA. Please add this result to the main text as it is relevant to addressing your first research question.
12. Line 381: Similar to Comment #10; heterotrophic picoplankton abundances are not shown in Figure 1. Please add in a reference to Supplemental Figure 5F.
13. Line 387: The phrasing is a bit confusing because Figure 2 appears to only show the AUV samples while Supplemental Figure 6 shows the AUV and CTD samples. Please be very clear throughout the results about which figures and analyses use AUV + CTD samples versus only the AUV samples.
14. Line 392: Please be a little more specific in word choice and change “most active” to “most active Cyanobacteria”.
15. Lines 400 – 401: Please state the average relative transcript expression for the eukaryotic genera listed in these lines.
16. Lines 406 – 407: Please state the average expression per transcript for the genera listed in these lines.
17. Lines 417 – 420: Please state which genera and KOs you’re referring to.
18. Lines 432 – 435: These KOs are not represented in Figure 2C, please add in the correct figure reference or state in the text that this is “data not shown”.
19. Line 436: Please state the KOs for ammonia oxidation and add in a reference to Figure 2 after “(Thaumarchaeota)”. This will help to distinguish between the data that is represented in Figure 2 versus Supplemental Figure 8.
20. Line 437: tonB-dependent receptor in AG-430-B22 is not shown in either Figure 2 or Supplemental Figure 8. Please either revise your figures to include it or remove it from the text.
21. Line 444: Change “fluorescence” to “chlorophyll concentrations”.
22. Line 445: Please define the hours that are “morning”, “afternoon”, “evening”, and “night” before using these terms to describe the results.
23. Line 445 – Line 449: Please explain why only data from leg 1 of the cruise was analyzed and represented in Figure 3A – 3C.
24. Line 465 – 467: As above for Comment #4.
25. Line 503: Please describe how the communities differ between leg 1 and leg 2. For example, reductions in the abundance of Cyanobacteria.
26. Line 503: It is unclear to me how Supplemental Figures 12 and 13 show differences in microbial community composition.
27. Line 508: Please add in a reference to Figure 4, and please make it clear in the text that *Nitrosopelagicus* is represented as Thaumarchaeota in Figure 4G.

Discussion:

28. Line 525: Please change “decay” to “weakening”.
29. Line 586: Please also include distance from eddy center as an important changing factor.

Figures:

30. Figure 1A: Please state in the figure caption what the arrows on the map are representing.
31. Figure 1B: Please state why a section of data is missing or omitted from this plot.
32. Figure 2A: Please state in the figure caption what “Time →” represents and how this corresponds to sampling location in the eddy.
33. Figure 4D: Please annotate the panels as leg 1 and leg 2.
34. Figure 4E: Please add the individual measurements of PC, PN, and PIC flux at Station ALOHA as points.
35. Supplemental Figure 1:
 - a. Please add panel labels to the figure.
 - b. Please state in the figure caption what the black circle is in the left panel.
 - c. Please add to the right panels, something that denotes when leg 1 and leg 2 of the cruise occurred. This will help the reader better interpret how the collected samples align with the process of eddy decay.
36. Supplemental Figure 2:
 - a. Please add panel labels.
 - b. Please add red dots to denote where transcriptomic samples were collected (like in Figures 1B and 1C)
37. Supplemental Figure 3:
 - a. Please make the shapes representing CTD and AUV sample locations larger.
 - b. Please add a boundary to the green shapes to make them more distinguishable.
38. Supplemental Figure 4:
 - a. Please add panel labels.
 - b. Please define what SLA_{corr} minimum is in the figure caption.
39. Supplemental Figure 5: Please increase the font of the R^2 and p-values.
40. Supplemental Figure 6: Please state in the figure caption how sample collection time corresponds with sample position in the eddy.
41. Supplemental Figure 9: It is not possible to see the diel expression of individual transcripts in this plot. Please choose different line colors or line types to help distinguish patterns of individual transcripts.
42. Supplemental Figures 10 and 11: It is unclear what the difference between these figures is.
43. Supplemental Figures 10, 11, and 12: Please state what the two different panels within the left panel represent. I suspect that they represent the two different legs of the cruise, but this needs to be clarified.

Supplemental Information:

44. Line 11; Lines 17 - 20: These lines of text highlight that there is an increase in sampling distance from the eddy center during leg 2 of the cruise and that this increase in distance influences both the water column properties and the biological patterns. This is a major result that affects the interpretation of the study. Please include the effect of distance from eddy center on the observed patterns in the main text.

45. Line 108: Add a comma after Cyanobacteria.
46. The supplemental information contains a lot of pertinent information for interpreting the results of this study. In the main text, please refer the reader more often to the relevant corresponding supplemental information.